JCB Journal of Cell Biology

# Giant worm-shaped ESCRT scaffolds surround actin-independent integrin clusters

Femmy C. Stempels[1], Muwei Jiang[1]*, Harry M. Warner[1]*, Magda-Lena Moser[1], Maaike H. Janssens[1], Sjors Maassen[1], Iris H. Nelen[1], Rinse de Boer[1], William F. Jiemy[2], David Knight[3], Julian Selley[3], Ronan O'Cualain[3], Maksim V. Baranov[1], Thomas C.Q. Burgers[4], Roberto Sansevrino[5], Dragomir Milovanovic[5], Peter Heeringa[6], Matthew C. Jones[7], Rifka Vlijm[4], Martin ter Beest[8], and Geert van den Bogaart[1,6]

Endosomal Sorting Complex Required for Transport (ESCRT) proteins can be transiently recruited to the plasma membrane for membrane repair and formation of extracellular vesicles. Here, we discovered micrometer-sized worm-shaped ESCRT structures that stably persist for multiple hours at the plasma membrane of macrophages, dendritic cells, and fibroblasts. These structures surround clusters of integrins and known cargoes of extracellular vesicles. The ESCRT structures are tightly connected to the cellular support and are left behind by the cells together with surrounding patches of membrane. The phospholipid composition is altered at the position of the ESCRT structures, and the actin cytoskeleton is locally degraded, which are hallmarks of membrane damage and extracellular vesicle formation. Disruption of actin polymerization increased the formation of the ESCRT structures and cell adhesion. The ESCRT structures were also present at plasma membrane contact sites with membrane-disrupting silica crystals. We propose that the ESCRT proteins are recruited to adhesion-induced membrane tears to induce extracellular shedding of the damaged membrane.

## Introduction

Endosomal Sorting Complex Required for Transport (ESCRT) proteins are critical membrane remodeling proteins found in all domains of life (Lindås et al., 2008; Samson et al., 2008; Liu et al., 2021). Whilst ESCRTs have long been known to regulate the formation of intraluminal vesicles in multivesicular bodies (MVBs; Katzmann et al., 2001), additional roles for ESCRT proteins keep being identified. For example, ESCRT proteins are also involved in the repair of the nuclear membrane (Denais et al., 2016; Raab et al., 2016) and the plasma membrane (Jimenez et al., 2014; Scheffer et al., 2014; Ritter et al., 2022). For plasma membrane repair, the membrane remodeling polymers of the ESCRT-III subfamily are proposed to trap the damaged part of the membrane on buds that are pinched off. These wound-closing ESCRT accumulations appear as scattered puncta of up to 2 µm in size at membrane regions damaged by a laser or pore forming agent (Jimenez et al., 2014; Scheffer et al., 2014). In COS-7 cells overexpressing ESCRT-III proteins, micrometer-sized spirals are formed at the plasma membrane (Cashikar et al., 2014). However, the formation of large ESCRT structures in non-manipulated cells in the absence of induced damage has not yet been shown.

Recent studies show that a sequential polymerization of different ESCRT-III proteins leads to membrane constriction (Nguyen et al., 2020; Pfitzner et al., 2020). In these studies, recruitment of the ESCRT-III protein IST1 is associated with an increase of the curvature of the membrane. However, the IST1-induced high curvature may be due to the process in which IST1 is recruited to pre-existing polymers, rather than it being an intrinsic characteristic of IST1. Notably, IST1 has also been shown to be involved in lower curvature polymers at the midbody of dividing cells (Bajorek et al., 2009; Guizetti et al., 2011). In addition, other ESCRT-III proteins, like CHMP4A, are shown to form low-curvature, flat spirals (Cashikar et al., 2014).

Cells adhere to the extracellular matrix (ECM) through a variety of cell-surface receptors, including integrins. The best-characterized cell-matrix adhesions are focal adhesions, which

[1]Department of Molecular Immunology, Groningen Biomolecular Sciences and Biotechnology Institute, University of Groningen, Groningen, The Netherlands; [2]Department of Rheumatology and Clinical Immunology, University of Groningen, University Medical Center Groningen, Groningen, The Netherlands; [3]Biological Mass Spectrometry Core Facility, Faculty of Biology, Medicine & Health, Manchester Academic Health Science Centre, University of Manchester, Manchester, UK; [4]Department of Molecular Biophysics, Zernike Institute for Advanced Materials, University of Groningen, Groningen, The Netherlands; [5]Laboratory of Molecular Neuroscience, German Center for Neurodegenerative Diseases, Berlin, Germany; [6]Department of Pathology and Medical Biology, University of Groningen, University Medical Center Groningen, Groningen, The Netherlands; [7]Peninsula Medical School, University of Plymouth, Plymouth, UK; [8]Department of Tumor Immunology, Radboud Institute for Molecular Life Sciences, Radboud University Medical Center, Nijmegen, The Netherlands.

*M. Jiang and H.M. Warner contributed equally to this paper.   Correspondence to Geert van den Bogaart: g.van.den.bogaart@rug.nl.

utilize integrin adaptors talin, paxillin, and vinculin to connect the ECM to the actin cytoskeleton (Horton et al., 2015). Focal adhesions connect to stress fibers, enabling the generation of contractile forces via myosin, for instance, for cell migration (as reviewed in Gardel et al. [2010]). Although by far most adhesions are connected to actin, actin-independent adhesions also exist. Examples are integrin αVβ5-containing reticular adhesions, which enable attachment during mitosis and respreading of the daughter cells (Lock et al., 2018), and clathrin-containing adhesions, which play a role in mechanosensitive signaling (Baschieri et al., 2018) and form anchoring points on collagen fibers (Elkhatib et al., 2017). Finally, hemidesmosomes are a type of actin-independent adhesion containing integrin α6β4, which connect the keratin cytoskeleton of epithelial cells to the ECM (Moch and Leube, 2021).

Here, we report clusters of micrometer-sized ring- and worm-shaped ESCRT structures at plasma membrane sites. The cortical F-actin cytoskeleton is absent at the position of the ESCRT structures, which is a hallmark of both membrane damage and extracellular vesicle formation (Miyake et al., 2001; Kalra et al., 2016). The ESCRT structures are devoid of ESCRT-interacting phosphoinositide lipids. The ESCRT structures surround membrane domains enriched in integrins and the known markers of extracellular vesicles tetraspanin CD63, ubiquitin (Buschow et al., 2005), and glycosylphosphatidylinositol (GPI)-anchored proteins (Vidal, 2020). The ESCRT structures are tightly adhered to the extracellular support, likely indirectly via the integrin-enriched clusters, and are left behind by the cell together with surrounding patches of plasma membrane. The ESCRT structures also form in the presence of the calcium chelator EDTA (which prevents integrin activation) when cultured on poly-L-lysine-coated supports, showing that integrin-independent adhesion to poly-L-lysine suffices for the formation of ESCRT structures. The ESCRT structures are broadly present in macrophages, fibroblasts, and dendritic cells. Immunohistochemistry on different human tissues shows that these cell types are highly positive for IST1. Although the structures form in the presence of serum in 3D collagen matrices, serum blocks their formation in cells cultured on glass. It is unlikely that starvation plays a role in the formation of the ESCRT structures, as coating of the glass support with serum or ECM proteins suffices for blocking their formation, and they form within minutes after seeding. As we show that the plasma membrane integrity is compromised in the absence of serum, we suggest that the ESCRT structures form in response to decreased plasma membrane integrity. This is further supported by the findings that the ESCRT structures are recruited to the plasma membrane contact sites with membrane disrupting silica crystals (Beckwith et al., 2020) and that the number of ESCRT structures increases upon loss of support by the cytoskeleton due to actin depolymerization.

Together, these data link the ESCRT structures to cell adhesion, membrane repair, and extracellular vesicle formation. Although it is challenging to assign specific cellular functions to the large ESCRT structures, because ESCRT proteins have widespread functions and it is not possible to only interfere with the ESCRT proteins within the structures, our evidence supports a model in which the structures repair adhesion-induced membrane damage. In addition, since live-cell imaging revealed that the ESCRT structures are highly immobile, they might have a scaffolding function in which they protect the plasma membrane.

## Results

### Worm-shaped IST1 structures at the plasma membrane

We discovered a new micron-sized structure at the plasma membrane. After culturing of human blood-isolated monocyte-derived dendritic cells (moDCs) on glass supports (Fig. 1 A) or in 3D collagen matrices (Fig. 1 B, Fig. S1, A and B; and Videos 1, 2, 3, and 4), we observed clusters of structures containing the ESCRT-III protein increased sodium tolerance 1 (IST1). The structures could also be visualized in live cells by overexpression of IST1 fused to GFP (Fig. 1, C and D). In collagen matrices, we observed the structures not only at the plasma membrane but sometimes (<5% of the cells) also at the nuclear membrane (Fig. 2).

The IST1 structures also contained the ESCRT-III protein charged multi-vesicular body protein (CHMP) 1B, which is an interaction partner of IST1 (Agromayor et al., 2009). Super-resolution stimulated emission depletion (STED) microscopy revealed heterogenous shapes of the structures, with the CHMP1B and IST1-positive filaments forming assemblies of ring- and worm-like shapes (Fig. 1 E and Fig. S1 C). While the single filaments could be up to ~5 µm in length, they clustered in areas with sizes up to about 20 µm in diameter (Fig. 1 A).

Correlative light electron microscopy (CLEM) revealed that the structures were present at the ventral plasma membrane (the part of the membrane facing the glass support) and the structures co-localized with membrane deformations (Fig. 1 F and Fig. S1 D). 3D STED microscopy showed that the structures were ~24 nm thick, as the axial full width at half maximum (FWHM) intensity was 24 nm larger compared to just plain dye on a coverslip (Fig. 1 G and Fig. S1 E). Experiments with the fluorescent lipophilic membrane marker Memglow confirmed that the structures are located at the plasma membrane (Fig. S1 F). This was further confirmed by total internal resonance fluorescence (TIRF) microscopy of IST1-GFP (Fig. 1 D).

The structures were already visible at early time points after seeding (<20 min; Fig. S1 C), and live-cell TIRF microscopy revealed that the structures formed within minutes after cell adhesion (Video 5). Time-lapse TIRF microscopy also showed that the structures are immobile during the ~3 h time span of the imaging (Video 6), suggesting that they tightly connect to the cellular support. In fact, the structures persisted after cell death induced by phototoxicity (Video 7). Expression levels of IST1 did not change after seeding (Fig. S1 G), suggesting that the formation of the structures did not depend on a transcriptional process.

We investigated the presence of other ESCRT proteins in the IST1 structures by immunolabeling (Fig. 3). All the ESCRT proteins that we assessed were present in the ESCRT structures: In addition to ESCRT-III protein CHMP1B (Fig. 1 E), CHMP4A was present, as well as the ESCRT disassembly protein vacuolar protein sorting 4 (VPS4; Fig. 3, A and B). Spastin, a microtubule

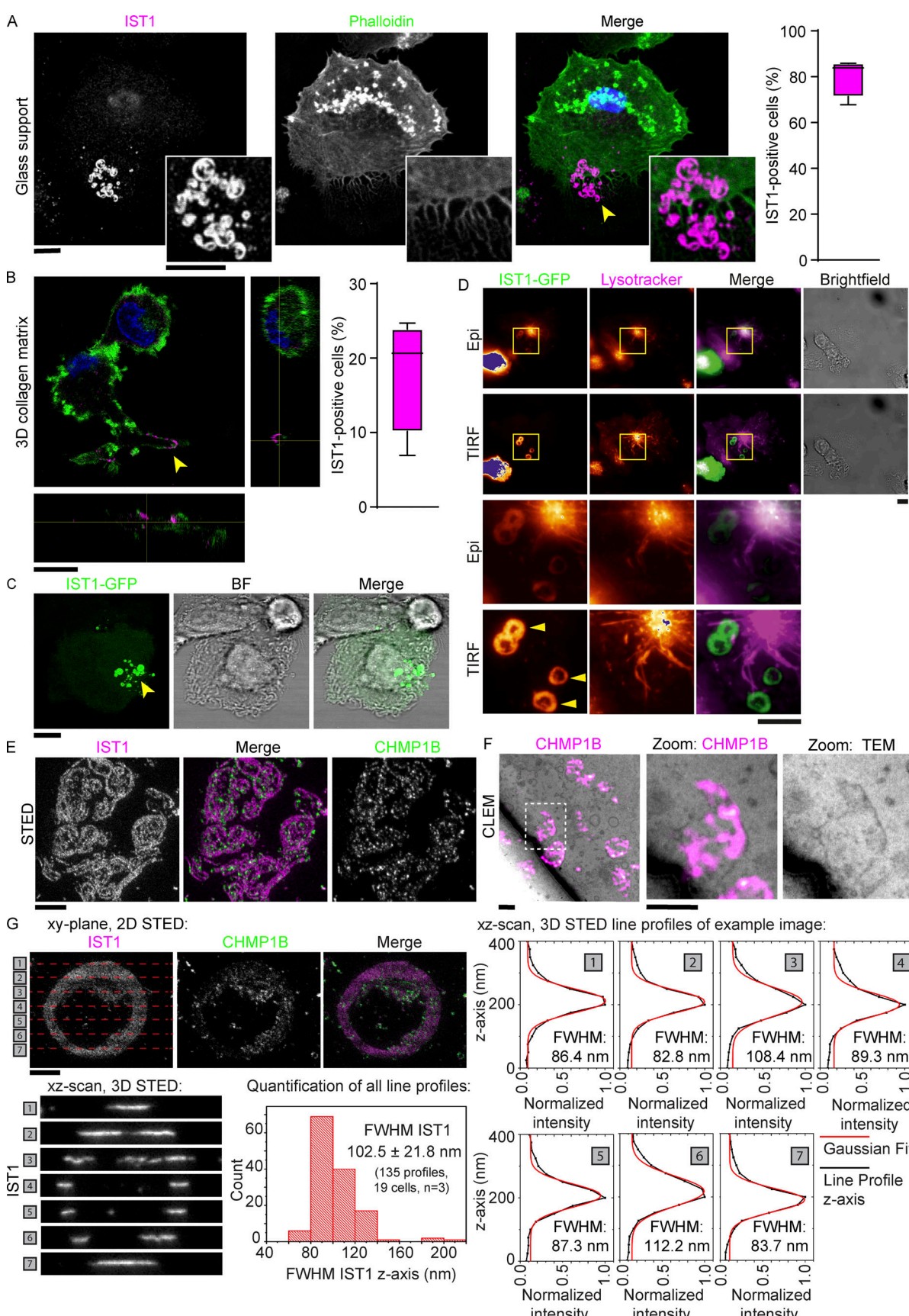

Figure 1. **IST1 and CHMP1B form clusters of flat ring- and tube-shaped structures at the plasma membrane of dendritic cells. (A)** Confocal micrograph of monocyte-derived dendritic cell (moDC) immunostained for IST1 (magenta in merge). Green: phalloidin. Blue: DAPI. The graph shows the percentage of cells

showing IST1 structures (*n* = 4 donors). Scale bars: 10 µm. **(B)** Confocal *z*-stack of moDC cultured in collagen matrix. The graph shows the percentage of cells showing IST1 structures (*n* = 4 donors). Scale bar: 10 µm. This image is part of Video 1. **(C)** Confocal micrograph of moDC overexpressing IST1-GFP. Scale bars: 10 µm. **(D)** Side-by-side comparison of conventional epi-fluorescence microscopy and total internal resonance fluorescence (TIRF) microscopy of moDC expressing IST1-GFP (green) and stained with lysotracker (magenta). Note the improved signal-to-noise of IST1 at the plasma membrane in TIRF. This image is part of Video 6. Scale bars: 10 µm. **(E)** STED microscopy of monocyte-derived dendritic cells (moDCs) cultured on glass supports immunostained for IST1 (magenta in merge) and CHMP1B (green). Scale bar: 1 µm. **(F)** Correlative light electron microscopy (CLEM) using fluoronanogold labeling of CHMP1B (magenta hot). TEM: transmission electron microscopy. Scale bars: 1 µm. **(G)** Top-left: 2D STED image of immunolabeled IST1 and CHMP1B structures in a moDC. The dashed red lines indicate the positions [1–7] at which single-line *xz*-scans were acquired using 3D STED (lower-left seven images). Of each of these *xz*-scans, the intensity distribution along the z-axis was obtained (right plots, black lines), and a Gaussian fit was made (lower plots, red lines) to determine the local thickness of the IST1 structure through the FWHM intensity of the Gaussian fits. In total, 19 cells from three donors were measured to obtain 135 line profiles. These results are shown in the lower-middle histogram, which shows an average FWHM of 102.5 ± 21.8 nm (±1 STD). Scale bar: 1 µm.

---

severing protein and known binding partner of IST1 (Agromayor et al., 2009), also co-localized with the IST1 structures (Fig. 3 A). ESCRT-I protein TSG101 was also present in the structures, as well as ALG-2 interacting protein X (ALIX) and the $Ca^{2+}$ adaptor protein ALG-2 that bridges ALIX and TSG101 (Okumura et al., 2009; Fig. 3 A).

ESCRT proteins have been shown to bind to 3-phosphoinositide lipids, e.g., the ESCRT proteins HRS and TSG101 bind to phosphoinositide 3-phosphate (PI(3)P; Raiborg et al., 2001; Teo et al., 2006), CHMP3 binds to phosphoinositide (3,5)-bisphosphate (PI(3,5)P$_2$; Whitley et al., 2003), and VPS36 binds to phosphoinositide (3,4,5)-triphosphate (PI(3,4,5)P$_3$; Slagsvold et al., 2005). However, two-color TIRF microscopy experiments with moDCs expressing mCherry-tagged CHMP4B (Jimenez et al., 2014) together with GFP-tagged pleckstrin homology (PH)-domain of phospholipase C delta (PLCδ; Stauffer et al., 1998), the PX-domain of p40phox (NCF4) specific for PI(3)P (Kanai et al., 2001), the N-terminal sequence of MCOLN1 specific for PI(3,5)P$_2$ (Li et al., 2013), or the PH-domain of AKT (Kwon et al., 2007), showed that the plasma membrane was devoid of phosphoinositide (4,5)-bisphosphate (PI(4,5)P$_2$), PI(3)P, PI(3,5)P$_2$, and PI(3,4,5)P$_3$ at the sites of the ESCRT structures (Fig. 4 A).

### ESCRT structures are present in migratory adhering cell types

In addition to the moDCs, we observed the ESCRT structures in human peripheral blood monocyte-derived macrophages, primary CD1c+ dendritic cells, and cultured primary human fibroblasts (Fig. 4 B). We also observed some structures in the murine macrophage-like cell line RAW 264.1, albeit in only a small fraction (<5%) of the cells (Fig. 4 B). Activation of the moDCs with the pathogenic stimulus lipopolysaccharide (LPS) did not affect the number of ESCRT structures (Fig. S2 A). The structures were not formed by blood-isolated monocytes (also not when stimulated with LPS), peripheral blood lymphocytes (PBLs), blood-isolated neutrophils, the kidney cell line HEK293, the alveolar epithelial cell line A549, and murine-cultured hippocampal neurons, astrocytes, and microglia (Fig. S2 B). Since we observed the phenotype in macrophages, dendritic cells, and fibroblasts, this argues for a role of the ESCRT structures in tissue infiltration. Indeed, immunohistochemistry of temporal artery biopsies of giant cell arteritis (GCA) patients showed bright IST1 labeling of CD68+ macrophages and vimentin+ fibroblasts (Fig. 5 A). We also observed IST1-positive infiltrating cells in other inflamed tissue (appendix; Fig. 5 B) and in migratory cell types in non-inflamed tissue (tonsil; Fig. 5 C).

### ESCRT structures do not colocalize with organellar markers

IST1 is associated with specialized ESCRT functions. In proliferating cells, ESCRT proteins play a role in cell division and IST1 locates to the fission ring (Agromayor et al., 2009), as we confirmed in HeLa cells (Fig. S3 A). However, moDCs are terminally differentiated cells that do not divide. IST1 also plays a role in the remodeling of endosomes (Allison et al., 2013) and has been shown to accumulate on endosomes in cells with mutated VPS4 (Rodger et al., 2020). We occasionally observed the late endosome marker LAMP1 in the centers of ring-shaped ESCRT structures (i.e., not overlapping but in the centers of the µm-sized rings; Fig. 6 A). In addition, we found that Sec22b, a Golgi SNARE protein, was occasionally present at the ESCRT structures (Fig. S3 B). However, we did not observe co-localization with the endosomal marker EEA1 (Fig. S3 C). We also did not observe co-localization of the ESCRT structures with markers for endoplasmic reticulum (PDI), mitochondria (TOMM20), the nuclear envelope (Lamin A/C), the ER-Golgi intermediate compartment (ERGIC53), the Golgi network (GM130), or autophagosomes (LC3; Fig. S3 D). These data suggest that the plasma membrane-localized ESCRT structures might originate from late endosomes and/or Golgi-derived vesicles.

### ESCRT structures might be involved in extracellular vesicle formation

The ESCRT machinery plays well-known roles in the formation of extracellular vesicles (Nabhan et al., 2012; Colombo et al., 2013; Juan and Fürthauer, 2018; Larios et al., 2020). For example, ALIX is known to sort tetraspanins (including CD63 and CD9) to extracellular vesicles (Larios et al., 2020). As we found that the composition of phospholipids is changed at the ESCRT structures, which is also the case in extracellular vesicles, we screened the ESCRT structures for extracellular vesicle markers. Indeed, in the centers of ring-shaped ESCRT structures (i.e., not overlapping with ESCRT), we occasionally observed the exosome cargoes CD63 and ubiquitin (Fig. 6 A). However, CD9 did not localize with the ESCRT structures (Fig. 6 A). We also overexpressed IST1-GFP with GPI-anchored RFP (Nadler et al., 2013; Fig. 6 B and Fig. S4 A), because extracellular vesicles carry multiple GPI-anchored proteins (Vidal, 2020). We frequently (~25%) observed approximately twofold enrichment of GPI-anchored RFP in the centers of the ESCRT rings compared to the plasma membrane outside the ESCRT structures. We did not detect caveolin 1, which is also involved in extracellular vesicle formation (Ni et al., 2020), in the ESCRT structures (Fig. S4 B).

A   Cell 1:
Phalloidin IST1 DAPI
Confocal micrographs with 0.65 µm z-distance

Max. Intensity z-projection

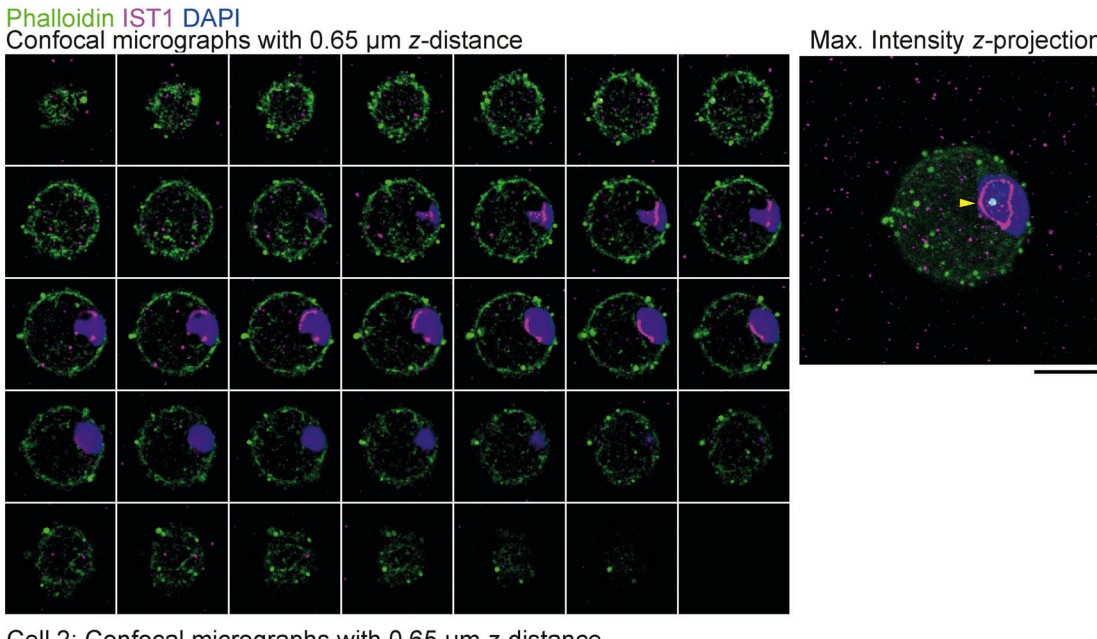

Cell 2: Confocal micrographs with 0.65 µm z-distance
Phalloidin IST1 DAPI

Max. Intensity z-projection

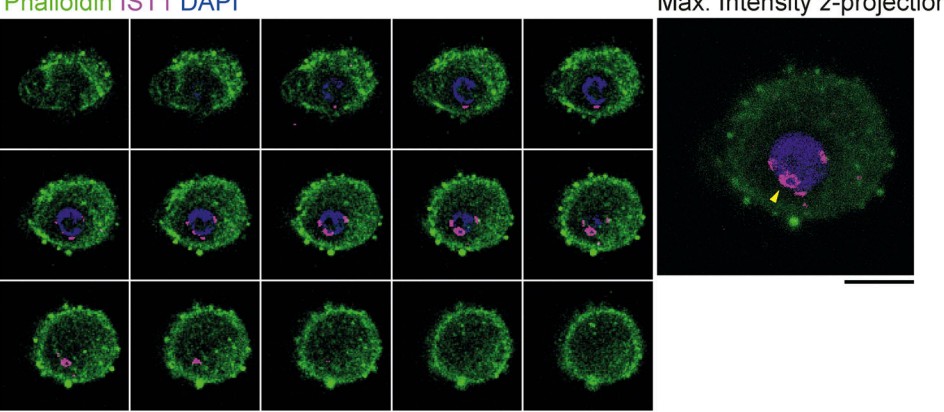

B   Cell 3:
IST1 DAPI Collagen

IST1 Collagen

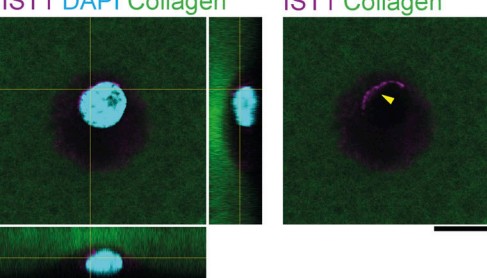

Figure 2.   **IST1 structures form at the nuclear membrane of cells cultured in collagen matrices. (A)** Monocyte-derived dendritic cells (moDCs) seeded on top of collagen matrix. Cells were allowed to migrate into the matrix for 5 h. Shown are confocal z-sections and maximum intensity z-projections. Cells are labeled for IST1 (magenta). Green: phalloidin. Blue: DAPI. **(B)** Same as A, but now with FITC-labeled collagen (green in merge) and no phalloidin. Shown are orthogonal sections. Scale bars: 10 µm.

The ESCRT system mediates exosome formation by forming intraluminal vesicles in multivesicular bodies. The fusion of these multivesicular bodies with the plasma membrane results in the release of these intralumenal vesicles (Colombo et al., 2013; Larios et al., 2020). ALIX is recruited to endosomes by the lipid lysobisphosphatidic acid, and ALIX in turn recruits ESCRT-III proteins to late endosomes, thereby promoting the sorting of ubiquitinated cargo molecules into the intraluminal vesicles (Larios et al., 2020). Overexpression of an mCherry-tagged ALIX truncation mutant (ALIXΔPRR), with the auto-inhibitory C-terminal proline-rich region (PRR) deleted, has been shown to increase exosome formation (Larios et al., 2020).

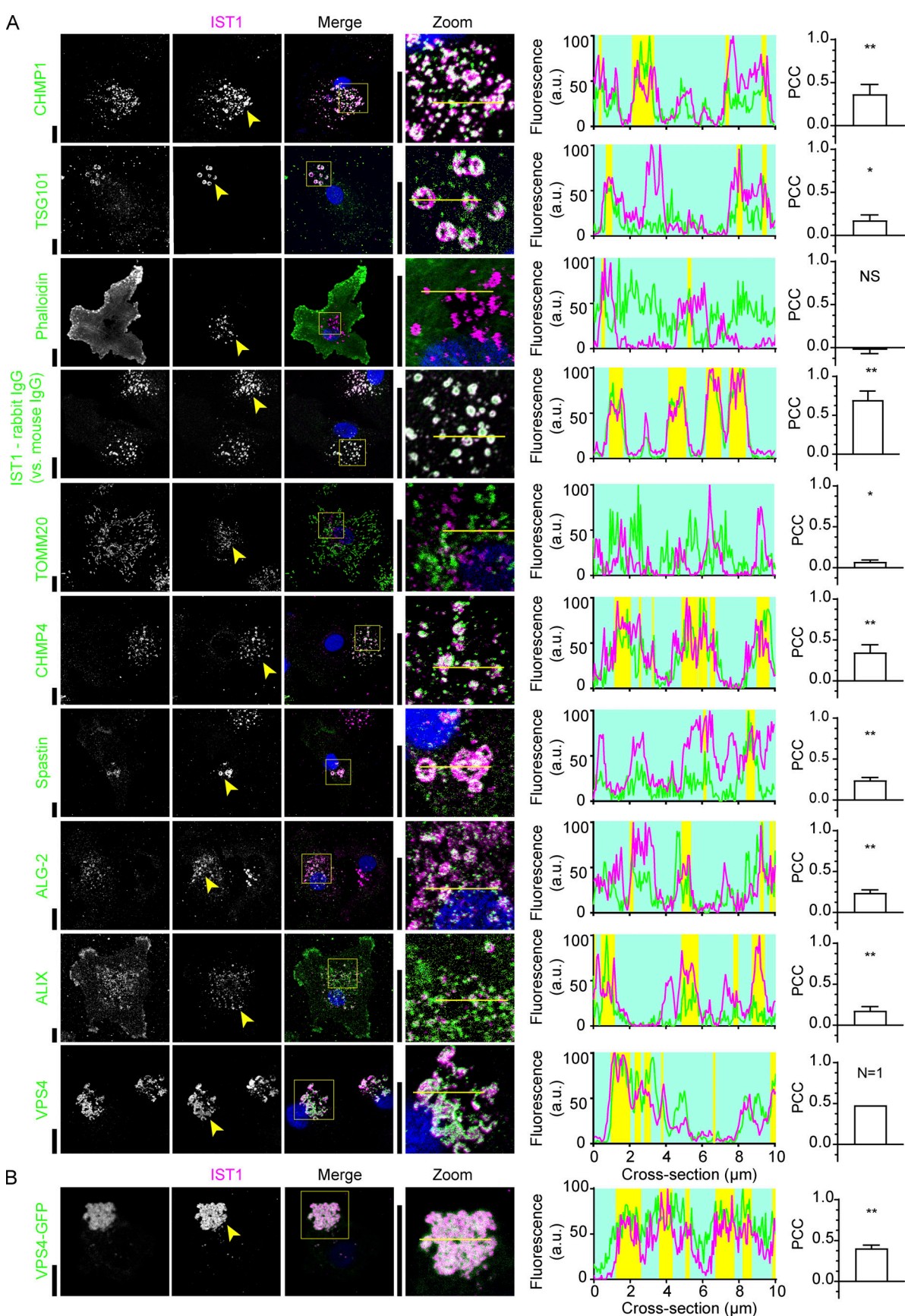

**Figure 3. IST1 structures contain other ESCRT components. (A)** Confocal micrographs of monocyte-derived dendritic cells (moDCs) immunolabeled for IST1 (magenta in merge) and the indicated proteins (green). Blue: DAPI. The line graphs show fluorescence intensity profiles as indicated by the yellow lines.

The yellow shaded areas indicate regions where the fluorescence intensity of both proteins exceeds the 50% intensity. The bar graphs show the PCC ± SD ($n \geq 3$ donors, except VPS4 for which we only succeeded to obtain $n = 1$; see B). PCC values were compared to 0 using one sample $t$ tests. Data distribution was assumed to be normal, but this was not formally tested. *: P < 0.05; **: P < 0.01; NS: not significant. **(B)** MoDCs transfected with VPS4-GFP and immunolabeled for GFP (green) and IST1 (magenta; $n = 3$ donors). Scale bars: 10 µm.

However, the ALIXΔPRR mutant was only expressed at very low levels compared to only mCherry and TIRF microscopy showed that it did not locate to the plasma membrane (Fig. S4 C). Moreover, ALIXΔPRR expression did not notably influence the ESCRT structures. It has previously been shown that this mutant is not recruited to the plasma membrane, likely because binding to lysobisphosphatidic acid is disrupted (Larios et al., 2020).

Another way how the ESCRT system can promote the formation of extracellular vesicles is by plasma membrane repair, where damaged membrane is shed into extracellular vesicle-like structures (Jimenez et al., 2014; Scheffer et al., 2014; Ritter et al., 2022). Indeed, starting from 1 h after seeding, the ESCRT structures were formed in patches of clustered structures that were no longer associated with the cells (Fig. 6 C). These extracellular clusters were still adhered to the glass support and were surrounded by plasma membrane, as shown by HLA-DR positive areas surrounding the clusters (Fig. 6 D). These data thus show that the entire clusters of ESCRT structures are released from the cells and argue for a role of the ESCRT structures for the formation of large extracellular vesicle-like structures.

### ESCRT structures depend on ECM

On glass supports, we noticed that the ESCRT structures were only formed when the cells were cultured in the absence of fetal bovine serum (FBS; Fig. 7 A). Moreover, replacing the medium of already adhered cells for medium lacking serum did not induce the formation of the structures, whereas reseeding the cells to fresh glass supports without serum did (Fig. 7 B). These findings suggest that the coating of the glass with serum components interferes with the formation of the structures. Serum contains ECM components, and deposition of serum-derived ECM molecules like fibronectin can condition the glass surface (Baier and Weiss, 1975). Indeed, the ESCRT structures were not formed on coverslips that were coated with non-polymerized collagen, fibronectin, or FBS (Fig. 7 A). However, even in the presence of serum, they were formed in 3D collagen matrices (Fig. 1 B and Fig. S1 A; and Videos 1, 2, 3, and 4), suggesting that ESCRT structures are not formed on unordered fibronectin or collagen coatings, but do form in matrices with fibrous collagen. Coating the glass supports with poly-L-lysine, which results in tight cell adhesion by electrostatic interactions, also did not interfere with the formation of the structures (Fig. 7 A). These findings indicate that the ESCRT structures are involved in strong cell adhesion to collagen fibers and non-specific electrostatic adhesion. Supporting this conclusion, the structures were left behind on the glass support upon cold-shocking the cells, which results in detachment of moDCs (Fig. 7C).

The finding that the structures are not formed in the absence of serum when the glass support is coated with collagen, fibronectin, or FBS, argues against a role for serum starvation in triggering the formation of the structures. In addition, the mTORC inhibitor rapamycin (50 nM) failed to induce the formation of the structures in the presence of FBS, further indicating that the formation of the ESCRT structures is not mediated by starvation. In addition, RPMI contains amino acids and the structures are formed within minutes after seeding (Fig. 6 C and Video 5); a period in which starvation is not yet expected.

### ESCRT structures surround integrin clusters

To identify other proteins in the ESCRT structures, we performed a pulldown of IST1 in cells seeded with and without serum followed by mass spectrometry (Fig. S5 A). STRING analysis identified proteins that are associated with cell adhesion, particularly integrins (Fig. S5 A). Indeed, we observed ESCRT spirals being wrapped around integrin clusters of integrin αM and β2 (i.e., no colocalization, but within the centers of the µm-sized ESCRT rings; Fig. 8 A). However, we do not believe that the ESCRT structures directly interact with the integrins, due to the large distance between the central integrin clusters and the surrounding ESCRT structures: this distance is >200 nm, given that we did not observe overlap in our microscopy with diffraction-limited resolution. Indeed, we could not detect interaction between immunoprecipitated IST1 and integrin β2 by Western blot (Fig. S5 B). As integrin (αMβ2, also called Mac-1) is known to be present in extracellular vesicles (Pluskota et al., 2008), this is in line with our model that ESCRT structures surround membrane domains enriched in cargoes for extracellular vesicles. For one out of the four tested donors, we also observed co-localization of ESCRT structures with integrin αVβ5 (Fig. S5 C), possibly due to interindividual variations in expression levels of this integrin.

The percentage of cells showing ESCRT structures that surrounded integrin clusters increased in the first 40 min after seeding and slightly decreased afterwards (Fig. 8 A). Immunohistochemistry of temporal artery biopsies of GCA patients showed co-expression of integrin β2 and IST1 in CD68[+] macrophages and vimentin[+] fibroblasts (Fig. 8 B).

In order to determine whether the formation of ESCRT structures depends on integrin-mediated cell adhesion, we seeded the cells on poly-L-lysine coated glass supports in the presence of 5 mM EDTA. EDTA is a calcium chelator and prevents activation of the calcium-dependent integrins. The ESCRT structures were still formed in the presence of EDTA, although the cells were rounded and less stretched and the number of the ESCRT structures was reduced in two out of three donors (Fig. 8 C). The overexpression of YFP-tagged integrin β2 (Kim et al., 2003) also did not increase the size and the number of the structures (compared to the YFP only control; Fig. S5 D). These data show that integrin-mediated cell adhesion is not required for the formation of the ESCRT structures. However, the ESCRT structures must be associated with cellular adhesion sites,

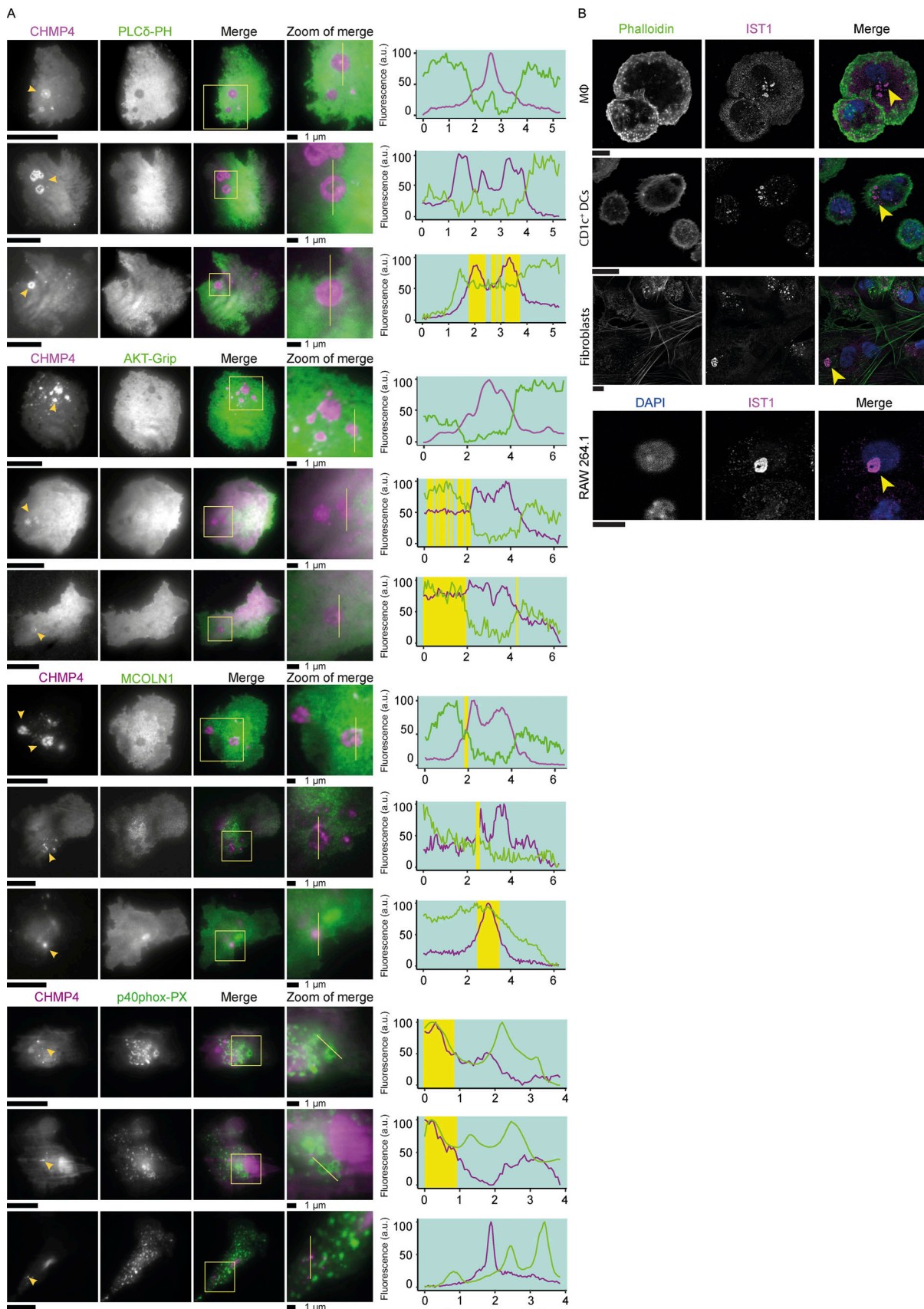

Figure 4. **IST1 structures are devoid of phosphoinositide (PI) lipids. (A)** Representative TIRF microscopy of moDCs co-expressing mCherry-tagged CHMP4 (magenta in merge) with GFP-tagged probes for PI lipids. The following PI probes were used: the PH-domain of PLCδ1 for PI(4,5)P₂, the PH-domain of AKT for

PI(3,4,5)P₃, the N-terminal sequence of MCOLN1 for PI(3,5)P₂, and the PX-domain of NCF4 (p40^phox) for PI(3)P (n = 3 donors). The line graphs show fluorescence intensity profiles as indicated by the yellow lines. The yellow shaded areas indicate regions where the fluorescence intensity of both proteins exceeds the 50% intensity. **(B)** Confocal micrographs of monocyte-derived macrophages (MΦ), CD1c⁺ dendritic cells, primary dermal fibroblasts, and mouse macrophage cell line RAW 264.1. Arrows indicate IST1 structures. Scale bars: 10 µm.

because: (i) in cells cultured on glass supports the ESCRT structures were only present at the ventral (i.e., adhering) side of the plasma membrane; (ii) the structures are immobile for multiple hours, (iii) the ESCRT structures are left behind by the cells upon migration and upon cold shock-induced detachment.

The mass spectrometry also revealed proteins involved in organelle trafficking and in immune signaling (Table S1). Next to the SNARE protein Sec22b (see above), immunofluorescence microscopy confirmed the presence of the calprotectin subunit s100a8 in the ESCRT structures (Fig. S6). We could not confirm colocalization of other proteins for which we had antibodies available (Fig. S6).

### ESCRT structures do not contain F-actin

Most ECM adhesion sites, such as focal adhesions and podosomes, are linked to the F-actin cytoskeleton (Horton et al., 2015). However, we did not observe F-actin at the ESCRT structures (Fig. 1 A), or at the central integrin clusters (Fig. 8 A). Tubulin was also not present at the structures (Fig. 9 A). In addition, the structures lacked the adapter proteins vinculin and talin that link integrins to the F-actin cytoskeleton (Fig. 9 B). Known F-actin-independent ECM adhesion structures are reticular adhesions (Lock et al., 2018) and clathrin-containing adhesion complexes (Elkhatib et al., 2017; Lock et al., 2019). However, although we observed integrin αVβ5 for one of the donors (Fig. S5 C), the ESCRT structures did not contain other marker proteins for reticular adhesions: NUMB, DAB2, nor clathrin light chain (Fig. 9 C). Importantly, the ESCRT-wrapped actin-independent adhesions can coexist with actin-containing adhesions like focal adhesions and podosomes (Fig. 9 B). These data show that the ESCRT structures surround integrin-independent cellular adhesion sites. Likely, this is related to the formation of extracellular vesicles and membrane repair, as local actin degradation is a hallmark of extracellular vesicle formation (Kalra et al., 2016) and membrane damage (Miyake et al., 2001).

### F-actin disruption increases the number of ESCRT structures and cell adhesion

To confirm that the ESCRT structures do not depend on the F-actin cytoskeleton, we performed experiments with the actin polymerization inhibitor latrunculin B. In the presence of this inhibitor, we no longer observed F-actin labeling but instead phalloidin gave a weak nuclear localization, confirming disruption of the F-actin cytoskeleton (Fig. S7 A). The depletion of consensus adhesion proteins (Horton et al., 2015) in response to latrunculin B treatment was further confirmed by mass spectrometry (Fig. 10 A and Fig. S7 B). However, surprisingly, the number of adherent cells increased almost twofold with latrunculin B (Fig. 10 B). At the same time, the number of ESCRT structures also increased upon treatment with this inhibitor (Fig. 10 B). This suggests that the cells shift toward actin-

independent adhesions in the absence of F-actin, and that these adhesions actually result in more cell adhesion than with regular, actin-dependent adhesions.

We hypothesized that the ESCRT structures would regulate cell attachment or detachment, e.g., by delivering integrins to the plasma membrane or by removing adhesions. To address this, we decreased expression levels of ESCRT proteins by siRNA knockdown and measured latrunculin B-induced changes in cell adhesion. Although siRNA knockdown does not allow us to discern specific functions of the ESCRT structures from other functions of the ESCRT proteins, such as multivesicular body formation, it is transient and therefore can be expected to limit cellular adaptation and does not interfere with the roles of ESCRT in cell division (Vietri et al., 2015). We first performed siRNA knockdown of IST1 and the upstream ESCRT-I protein TSG101 (for both 50–75% knockdown efficiency; Fig. S7 C). IST1 depletion decreased the sizes of the IST1 structures, confirming successful knockdown (Fig. S7 D). Despite this, knockdown of IST1 and TSG101 both increased cell adhesion after latrunculin B treatment for four out of five donors, suggesting that the ESCRT structures might work against adhesion (e.g., by removing the adhesions). However, the fifth donor showed the opposite effect (Fig. S7 E).

To further determine whether the ESCRT structures are linked to the formation of the actin-independent adhesions, we also investigated the effect of ESCRT knockdown on integrin β2 levels at the integrin clusters. Knockdown of IST1 or TSG101 both resulted in a small but significant reduction of the integrin β2 signal in the clusters surrounded by the ESCRT structures (Fig. S7 F). However, these data should be interpreted with caution as this effect likely relates to roles of ESCRT in endosomal trafficking and/or degradation of integrins, because flow cytometry showed that knockdown of TGS101 also significantly decreased the total surface exposure of integrin β2 in non-adhered cells (i.e., independent of the ESCRT structures; Fig. S7 G). Further arguing against a role of the ESCRT structures in the formation of the actin-independent adhesions is the finding that not all integrin clusters were surrounded by ESCRT structures (Fig. 8 A). In addition, the ESCRT-wrapping of the integrin clusters peaked at 40 min after seeding, whereas the ESCRT structures were visible within minutes, making it unlikely that ESCRT structures deliver integrins to the plasma membrane.

Thus, although the disruption of the F-actin cytoskeleton increases cell adhesion concomitant with more ESCRT structures, the ESCRT structures do not seem to be responsible for cellular adhesion and we found no evidence that the ESCRT structures regulate the formation or the removal of the actin-independent integrin clusters.

### ESCRT structures might repair the plasma membrane

ESCRT proteins mediate repair of laser-induced or toxin-induced plasma membrane damage (Jimenez et al., 2014;

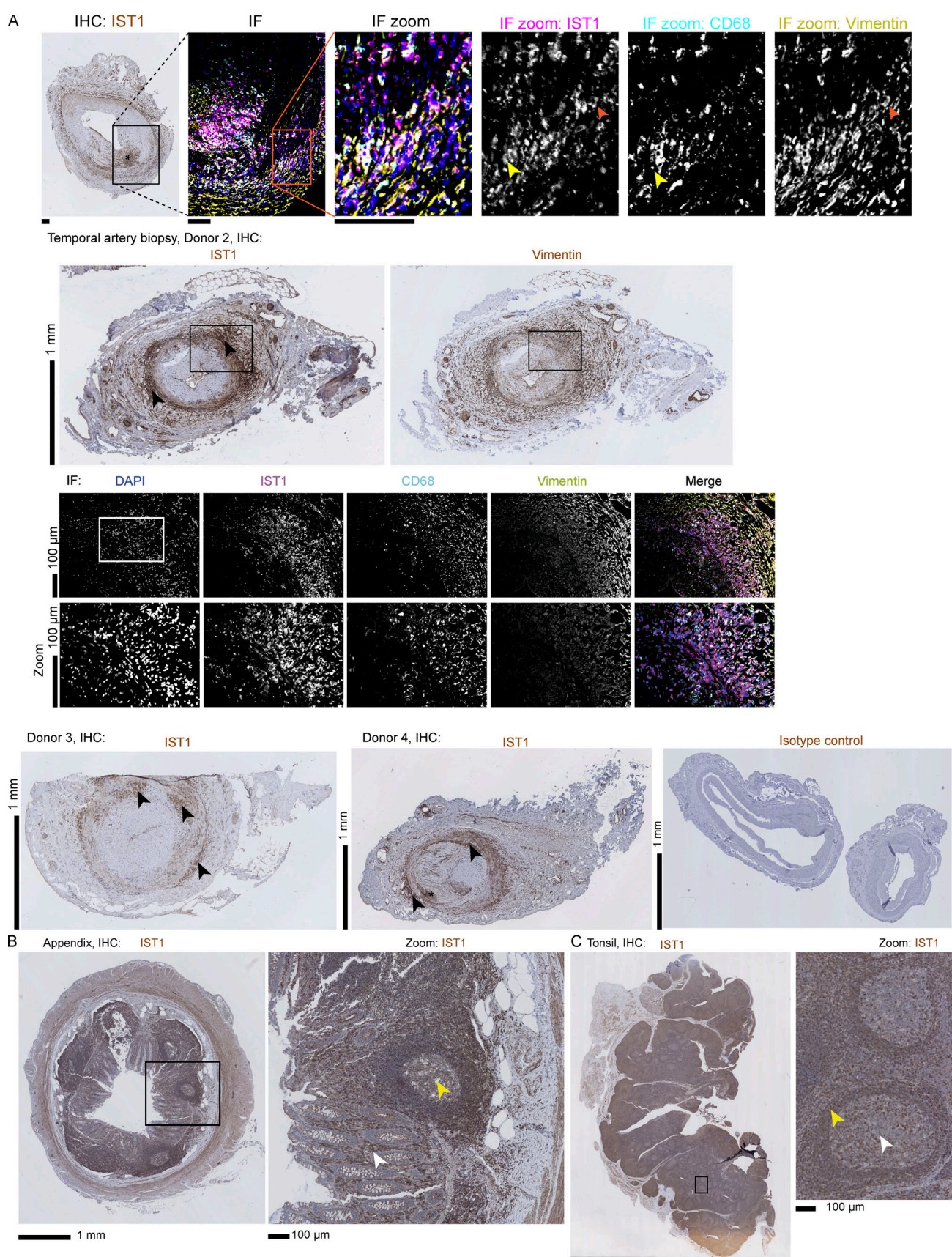

Figure 5. **IST1 structures are present in migratory adhering cell types. (A and B)** Immunohistochemistry (IHC) and immunofluorescence (IF) micrographs of temporal artery biopsies of four donors (A) and IHC of appendix and tonsil tissue of one donor (B). IHC sections are labeled for IST1, except for the right

image of the second donor of the temporal artery tissue; here an adjacent section was labeled for the fibroblast-marker vimentin. The IF images show the area indicated with a black rectangle in an adjacent section labeled for DAPI (blue), IST1 (magenta), macrophage-marker CD68 (cyan), and vimentin (yellow). The zoomed images show the area indicated with the white rectangle. Arrows indicate IST1-positive cells also positive for CD68 (yellow) or vimentin (orange). As a control for a specific binding of the IST1 antibody, an isotype control staining using a mouse IgG2a antibody was performed. In giant cell arteritis, macrophages infiltrate the adventitia of the vessel wall (black arrows). An inflammatory infiltrate breaking through the internal elastic lamina is labeled with an asterisk. Endothelial cells (e.g., of the vasa vasorum) are positive for IST1, too. The appendix shows the lamina propria (white arrow) in which macrophages and mostly plasma cells are positive for IST1. A lymphatic nodule is indicated with a yellow arrow. In the tonsil, stromal cells are positive for IST1. In the germinal centers (white arrow), plasma cells are positive for IST1 where macrophages are not. This might be explained by the low level of migration of macrophages in the germinal center. In the mantle zones (yellow arrow), macrophages are positive for IST1.

Scheffer et al., 2014). A potential mechanism of how latrunculin B treatment increased the presence of the ESCRT structures, is that this might increase the recruitment of ESCRT proteins to sites of membrane damage, as membrane resealing is inhibited when actin depolymerization is blocked and enhanced when it is increased (Miyake et al., 2001). In line with the known membrane-repairing roles of ESCRT proteins is the finding that we not only observed them at the plasma membrane, but also at the nuclear membrane in a small fraction (<5%) of cells cultured in collagen matrices (Fig. 2). In tumor cells, nuclear deformation caused by confining microenvironments such as dense collagen matrices is known to damage the nuclear membrane, and ESCRT proteins are involved in the repair of the nuclear membrane (Denais et al., 2016; Raab et al., 2016).

To test the presence of membrane damage, we compared the leakage of (membrane-impermeable) DAPI from the medium through the plasma membrane into cells cultured in the presence (no ESCRT structures are formed) versus in the absence of serum (ESCRT structures are formed). DNA staining dyes have been used previously as membrane leakage markers (Jimenez et al., 2014; Corrotte et al., 2012; Lam et al., 2019; Ritter et al., 2022). Indeed, we found more leakage of DAPI into cells that were cultured in the absence of serum (Fig. 10 C). This shows that the serum-free condition is indeed more challenging for the plasma membrane than the serum condition and suggests that the ESCRT structures are formed to cope with this challenging environment. The alternative (opposite) conclusion is that the structures are the cause of the leakage in the serum-free condition, but this is not in line with the functions of ESCRT proteins as described in literature. However, knockdown of TSG101 in cells cultured in serum-free conditions only increased DAPI influx for two out of three donors and had the opposite effect in the third donor (Fig. S7 H). Possibly, the levels of TSG101 were not reduced enough to hamper its function. We were unable to perform double knockdown of TSG101 with ALIX.

To overcome this challenge and directly determine whether the ESCRT structures formed at sites of membrane damage, we exposed the cells to silica crystals that are known to induce membrane damage (Beckwith et al., 2020). We observed large ESCRT structures containing IST1, CHMP4A, CHMP4B, and ALIX at the plasma membrane contact sites with silica crystals (Fig. 10 D). Live-cell imaging showed that IST1-GFP was also recruited to these contact sites (Fig. 10 D). Together, these findings suggest that the ESCRT structures mediate membrane repair of damage occurring at or close to the position of the actin-independent adhesion sites, and this leads to shedding of plasma membrane fragments.

## Discussion and conclusion

In this study, we found that macrophages and other migratory cell types form actin-independent adhesions that are enriched in markers for extracellular vesicles and surrounded by large worm-shaped ESCRT structures. The surrounding ESCRT structures differ from previously identified ESCRT structures as they form clusters of distinctive worm-shaped structures, are extremely large (up to 5 μm), and remain stably present at the plasma membrane. The structures are present in non-manipulated cells, as opposed to the spirals formed at the plasma membrane of CHMP4A-overexpressing or VPS4-depleted cells (Cashikar et al., 2014), and they form in physiological collagen matrices. Surprisingly, the ESCRT structures have not been observed previously, although they are very large and have distinctive ring-shaped and spiral-shaped morphologies. One possible explanation for this is that ESCRT proteins have been mainly studied in cell lines that do not form these structures, as we show that they are only formed in tissue-infiltrating cell types (macrophages, dendritic cells, and fibroblasts).

A question is why cells form actin-independent adhesions. Actin-dependent cell adhesions connect to the F-actin cytoskeleton to enable mechanosensing, cell protrusion, and migration within tissues. In contrast, the actin-independent integrin clusters might enable interactions of regions of the cell with the ECM for maintaining cellular shape and prevent the cells from being displaced, e.g., due to contractile or expansive forces resulting from tissue motion. Thus, although an unproven hypothesis, the actin-free integrin clusters might allow cells to stably remain in place within tissues.

The disruption of the F-actin cytoskeleton and the depletion of actin-dependent cellular adhesions by Latrunculin B surprisingly increased cellular adhesion, and this was accompanied by the increased abundance of the ESCRT structures. However, our data indicate that the ESCRT structures are not involved in actin-independent cell adhesion nor in the regulation of the actin-independent adhesion sites, because of three reasons: First, not all integrin clusters were surrounded by ESCRT structures, and the structures did not precisely locate at the sites of the integrin clusters, but were mostly surrounding them, and this surrounding was quite non-uniform: sometimes, we observed ring- or worm-shaped ESCRT structures surrounding multiple integrin clusters. Second, the ESCRT structures were also formed upon adhesion to poly-L-lysine substrates in the presence of an excess of the calcium-chelator EDTA, showing that their formation does not require integrin-dependent cell adhesion. Third, the ESCRT structures were visible almost immediately after seeding, whereas their wrapping of the integrin

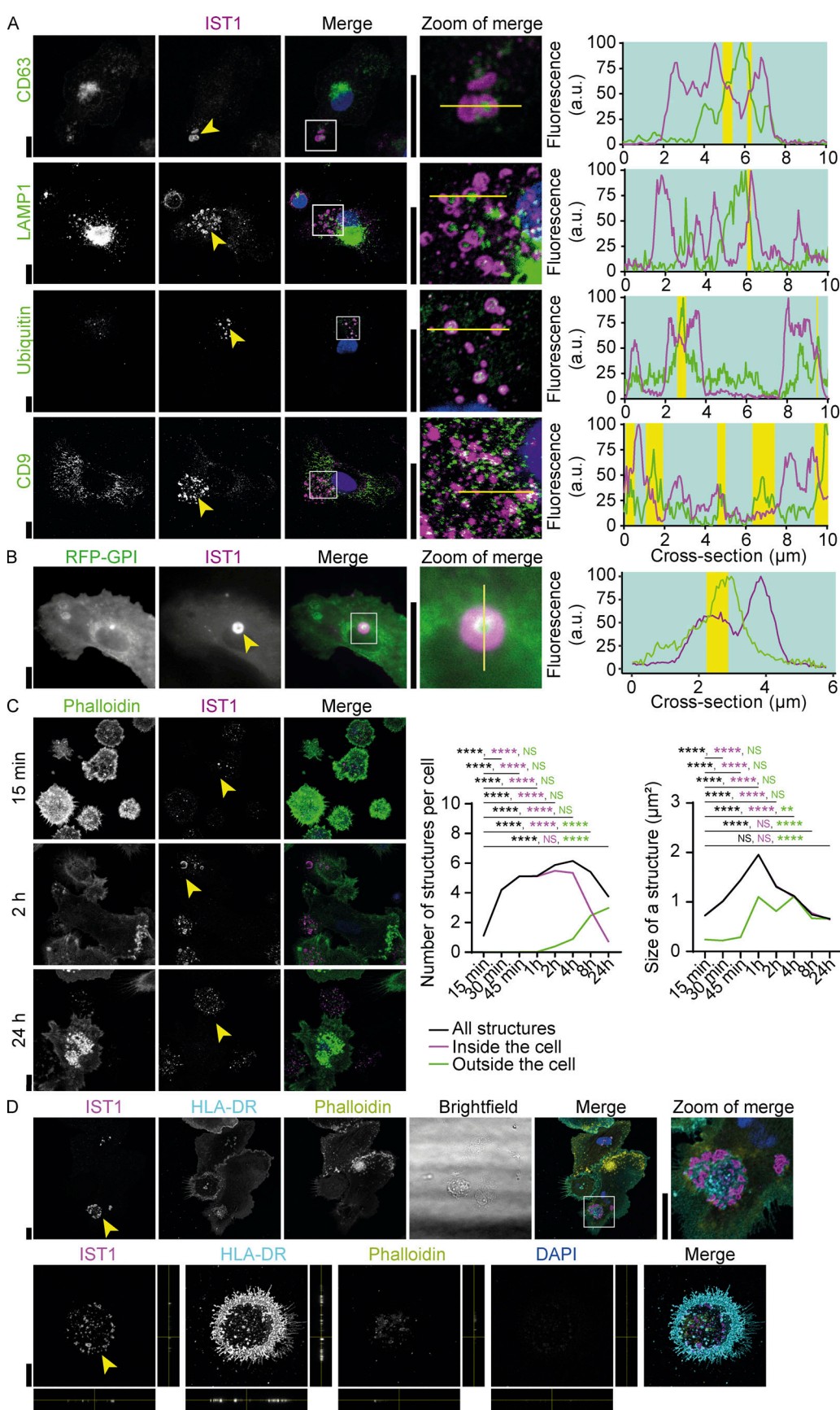

Figure 6. **IST1 structures surround markers of extracellular vesicles and are left behind by the cells. (A)** moDCs immunolabeled for IST1 (magenta in merge) and CD63, CD9, LAMP1, or ubiquitin (green). Blue: DAPI. Line graphs show fluorescence intensity profiles as indicated by the yellow line. Yellow shaded

areas show regions where the fluorescence intensities for both proteins exceed the 50% intensity. **(B)** TIRF microscopy of moDCs co-expressing GPI-anchored RFP (green) with GFP-labeled IST1 (magenta; $n$ = 3 donors). Scale bar: 5 µm. **(C)** Confocal micrographs of moDCs cultured for the indicated times and immunolabeled for IST1 (magenta) and phalloidin (green). Yellow arrowheads: IST1 structures inside (15 min) and outside (2 and 24 h) the cell. Left graph: average number of IST1 structures per cell localized inside or outside the cell (as determined by phalloidin; $n$ = 4 donors). Only structures with a surface area >0.3 µm² were included in this analysis. Right graph: average size of the structures. Time points are compared to the first time point (15 min) using a two-way ANOVA followed by Dunnet's post hoc test. Data distribution was assumed to be normal, but this was not formally tested. **: $P < 0.01$; ****: $P < 0.0001$; NS: not significant. **(D)** Top: Confocal micrographs of moDCs immunolabeled for IST1 (magenta), HLA-DR (cyan), phalloidin (yellow), and DAPI (blue). Bottom: Same staining of patches of left behind IST1-positive structures, including orthogonal views. Yellow lines indicate where the cross-section was taken. Scale bars: 10 µm unless indicated otherwise.

clusters peaked much later (40 min). Therefore, it is unlikely that the ESCRT structures deliver or remove integrins to or from the plasma membrane. The ESCRT structures rather seem a downstream consequence of the formation of adhesions.

We showed that knockdown of ESCRT proteins lowered surface levels of integrins independent of the ESCRT structures, as we showed that this lowered total surface levels of integrin β2 in non-adherent (and thus ESCRT structure-free) cells. These effects on integrin levels are likely caused by the roles of ESCRT in endosomal trafficking and/or multivesicular body formation. Overall, this shows the difficulty in assigning specific cellular roles to the ESCRT structures, as any perturbation of ESCRT proteins will also affect other functions of the ESCRT machinery.

Our data combined with literature suggest that the ESCRT structures are involved in the production of extracellular vesicles and in membrane repair (Fig. 10 E). The ESCRT structures surrounded clusters of known exosomal cargoes CD63, ubiquitin (Buschow et al., 2005), and GPI-anchored proteins (Vidal, 2020) and membrane patches containing the ESCRT structures are left behind by the cells and remain adhered to the substrate. Moreover, we observed large ESCRT structures at the plasma membrane contact sites with membrane damaging silica crystals, in line with previously reported recruitment of ESCRT proteins to sites of membrane damage induced by a laser or pore-forming agent (Jimenez et al., 2014; Scheffer et al., 2014). We also observed the large ESCRT structures at nuclear membranes of a small fraction of DCs cultured in collagen matrices. ESCRT proteins repair the nuclei of migrating tumor cells in confining microenvironments (Denais et al., 2016; Raab et al., 2016). Finally, we found increased formation of the ESCRT structures in the absence of serum where the plasma membrane is more permeable to DAPI. However, a caveat of our study is that all this evidence is mostly indirect, and the knockdown of ESCRT proteins showed no consistent effect on membrane leakage. As described above, knockdown of ESCRT proteins will not only affect the ESCRT structures, but also affects the trafficking of integrins and likely also interferes with other functions of ESCRT proteins. Therefore, it still needs to be directly tested whether ESCRT structures indeed mediate extracellular vesicle formation and/or membrane repair.

Finally, as we observed that the ESCRT structures are extremely stable and persist even after cell death and cell de-adhesion, the ESCRT structures might also have a scaffolding function and support regions of vulnerable membrane. The ESCRT structures we report in this study might resemble membrane-protective carpets that are formed by inner membrane-associated protein of 30 kD (IM30), an ESCRT-III

protein found in bacteria and plants, as shown by electron microscopy and a decreased proton flux through isopropanol-treated liposomes in the presence of this protein (Junglas et al., 2020; Liu et al., 2021). We therefore hypothesize that the ESCRT structures might function not only in the repair but also as similar supportive filaments for regions of vulnerable membrane.

## Materials and methods

### Patient materials
Four inflamed temporal artery biopsy tissue samples from giant cell arteritis patients were included. The study was approved by the institutional review board of the University Medical Center Groningen (METc2010/222), and written informed consent was obtained. Tonsil and appendix tissues were obtained as products of routine tonsillectomy and appendectomy. All procedures were conducted in compliance with the Declaration of Helsinki.

### Immunohistochemistry (IHC)
Formalin-fixed, paraffin-embedded tissues were cut into sections of 3 µm. The sections were deparaffinized and rehydrated, followed by antigen retrieval with tris-EDTA buffer (pH 9) for 15 min in a microwave. Tissues were incubated with primary anti-human IST1 (66989-1-Ig, 1:1,000 dilution; Proteintech) for 1 h at room temperature (RT), followed by endogenous peroxidase blocking with 3% H2O2 for 30 min. The tissues were subsequently incubated with EnVision anti-mouse peroxidase conjugated secondary antibody (DAKO), 3-amino-9-ethyl-carbazole (DAKO) for peroxidase activity detection, and finally hematoxylin (Merck) as a counterstain. All slides were scanned using a Nanozoomer Digital Pathology Scanner (NDP Scan U 10074-01; Hamamatsu Photonics). Images were analyzed using FIJI-ImageJ (Schindelin et al., 2012).

### Fluorescence multispectral tissue imaging
Paraffin sections were deparaffinized and rehydrated, followed by antigen retrieval in Tris-EDTA buffer (pH 9) for 15 min in a microwave. Next, tissues were incubated with a cocktail of primary antibodies (Table 1). Subsequently, they were incubated with a cocktail of secondary antibodies, followed by incubation with a cocktail of tertiary antibodies tagged with fluorescence labels. Autofluorescence was blocked with Vector TrueVIEW autofluorescence quenching kit (Vector laboratories) for 5 min according to the manufacturer's instructions. Afterwards, the tissues were incubated with DAPI for 10 min as counterstain and sealed. Image cubes were captured at a magnification of 20×

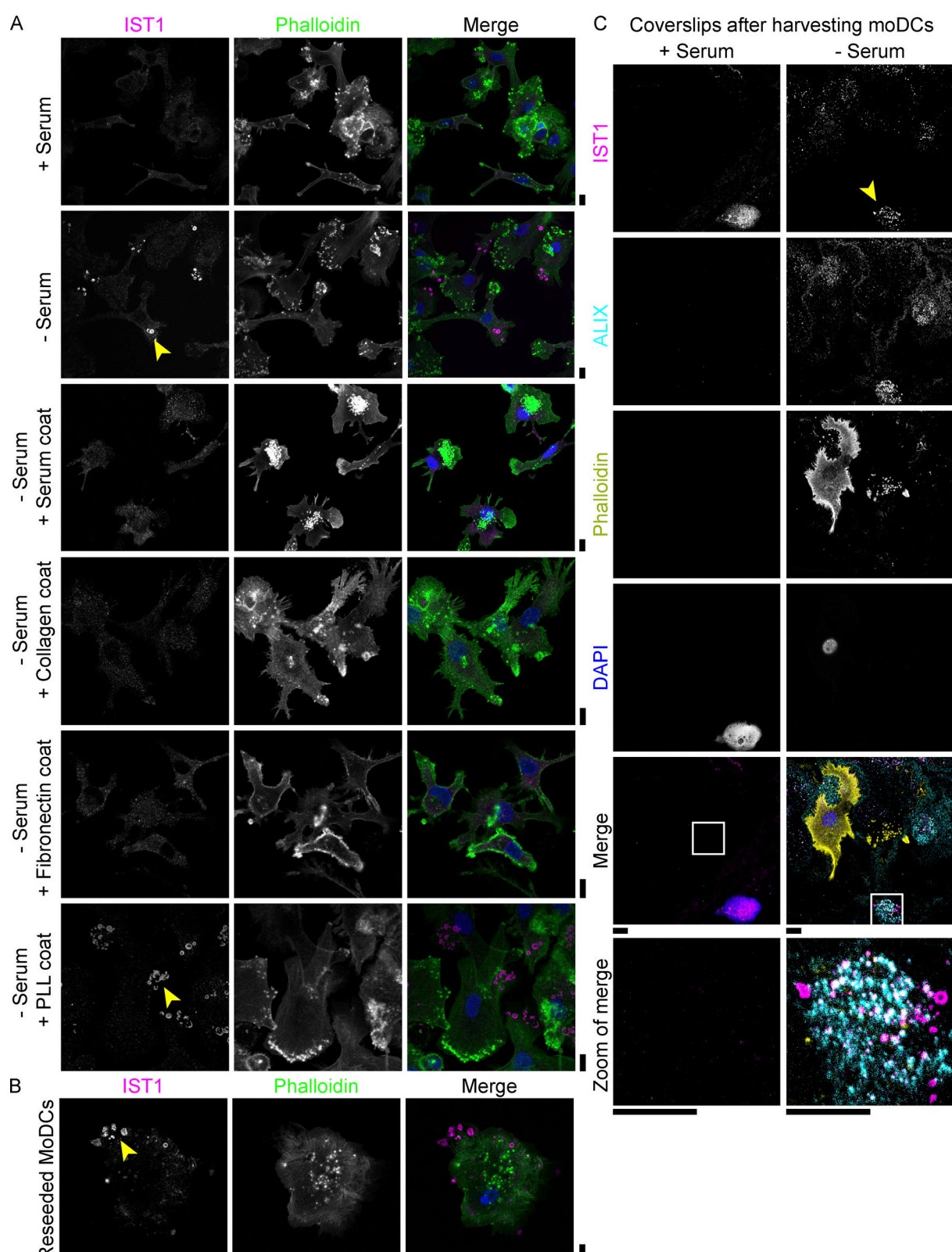

Figure 7. **IST1 structures form in absence of serum and tightly adhere to the cell substrate. (A)** Confocal micrographs of monocyte-derived dendritic cells (moDCs) cultured in the presence or absence of 10% fetal bovine serum, on glass coverslips that are either uncoated, coated with rat tail collagen, fibronectin, or poly-L-lysine (PLL). Yellow arrowheads indicate IST1 structures. **(B)** MoDCs cultured in the presence of serum were harvested with cold-procedure and subsequently reseeded on clean coverslips in the absence of serum. Confocal micrograph of reseeded moDC immunolabeled for IST1 (magenta), phalloidin (green), and DAPI (blue). **(C)** MoDCs were cultured overnight on coverslips in the absence or presence of serum, and cells were subsequently harvested using cold-procedure. Confocal micrographs of residue on the coverslips. To ensure imaging at the right z-plane, areas with a remaining cell (i.e., with a nucleus) are shown. Scale bars: 10 μm.

Figure 8. **ESCRT structures surround actin-independent integrin clusters. (A)** Confocal micrographs of monocyte-derived dendritic cells (moDCs) incubated for 40 min and immunolabeled for IST1 (magenta in merge), integrin αM or integrin β2 (cyan), and phalloidin (yellow). Blue: DAPI. Bar graphs show cells with integrin-wrapping IST1 structures as a percentage of all cells forming IST1 structures, at the indicated time points, for integrin β2 and integrin αM. Line graphs show fluorescence intensity profiles as indicated by the white line. ($n \geq 3$ donors; two-sided paired $t$ tests; *: $P < 0.05$; **: $P < 0.01$; NS: not significant). Scale bars: 10 µm. **(B)** Immunofluorescence labeling of a temporal artery biopsy (same donor 1 as in Fig. 5 A) for DAPI (blue), IST1 (magenta),

integrin β2 (yellow), and macrophage marker CD68 or fibroblast marker vimentin (cyan). **(C)** Confocal micrographs of moDCs seeded in the presence or absence of 5 mM EDTA, and immunostained for IST1 (magenta), integrin β2 (green), and phalloidin (yellow). Arrows: IST1-positive structures. Graph: average number of IST1-positive structured per cell ($n$ = 3 donors). For the statistical analysis of A and C, data distribution was assumed to be normal, but this was not formally tested. Scale bars: 10 μm.

using Nuance Multispectral Imaging System 3.0.1 (PerkinElmer) using NuanceFX 3.0.1 software (PerkinElmer). Filters used were 440:460 for DAPI, 490:530 for Alexa Fluor 488, 570:600 for Alexa Fluor 568, and 710:720 for Alexa Fluor 647. Spectral unmixing was performed with spectral libraries of each fluorophore subtracting the background signal.

### Primary cells

#### Isolation of monocytes, peripheral blood lymphocytes, and neutrophils, and generation of monocyte-derived dendritic cells and macrophages

Experiments were performed using human monocyte-derived dendritic cells (moDCs) unless stated otherwise. Approval to conduct experiments with human blood samples was obtained from the blood bank, and all experiments were conducted according to national and institutional guidelines. Informed consent was obtained from all blood donors by the Dutch blood bank. Samples were anonymized and none of the investigators could ascertain the identity of the blood donors. MoDCs were obtained from healthy donors (Baranov et al., 2016). Briefly, peripheral-blood mononuclear cells (PBMCs) were isolated from buffy coats by density gradient centrifugation. Peripheral blood lymphocytes (PBLs) were separated from monocytes by allowing the monocytes to adhere to the culture flask followed by extensive washing. PBLs and monocytes were either frozen for later use, as described below, or monocytes were differentiated into moDCs by culturing in RPMI 1640 medium supplemented with L-glutamine (21875-034; Gibco), 10% fetal bovine serum (FBS; 10309433; Thermo Fisher Scientific), 1% antibiotic-antimitotic (AA; 15240062; Gibco), 300 U/ml interleukin-4 (130-093-924; Miltenyi), and 450 U/ml granulocyte-macrophage colony-stimulating factor (130-093-867; Miltenyi) for 6 d at 37°C and 5% CO$_2$. Then, the moDCs were harvested and stored in liquid nitrogen in RPMI supplemented with 10% DMSO and 40% FBS. For the generation of monocyte-derived macrophages, CD14$^+$ monocytes were isolated using a CD14 Microbead kit (130-114-976; Miltenyi) according to the manufacturer's instructions and cultured with 100 ng/ml macrophage colony-stimulating factor (216-MC; Bio-Techne). Macrophages were not frozen but used for experiments immediately. Neutrophils were isolated from the bottom fraction of the buffy coat after density gradient centrifugation. This fraction, containing neutrophils and erythrocytes, was twice resuspended in 155 mM ammonium chloride and incubated on ice for 5 min, to lyse the erythrocytes. Neutrophils were washed and immediately used for experiments.

#### Isolation of CD1c$^+$ dendritic cells

CD1c$^+$ dendritic cells were isolated from PBMCs using the CD1c (BDCA-1)$^+$ dendritic cell isolation kit (130-119-475; Miltenyi) according to the manufacturer's instructions.

#### Primary fibroblasts culture

Human primary dermal fibroblasts were obtained from a healthy donor and maintained in DMEM (11960044; Gibco) supplemented with 1% glutamine (25030149; Gibco) and 1% AA.

#### Primary cultures of brain cells

Hippocampal neurons, astrocytes, and microglia were prepared from P0 wild-type mice (C57BL6/J). Hippocampi and cortexes were manually dissected from the whole brain and were separately collected in cold Hanks' balanced salt solution (HBSS, Gibco) containing 10 mM HEPES buffer (Gibco), 1 mM pyruvic acid (Gibco), 50 U/ml penicillin/streptomycin (Gibco), and 5.8 mM magnesium chloride. Briefly, for neurons, hippocampi were enzymatically digested with Papaine (Sigma-Aldrich) in HBSS for 20 min at 37°C. Papaine was removed with repeated HBSS washing, and plating medium was added (Neurobasal Medium-A [Gibco], supplemented with 5% FBS [FACS], 1% B27 [Gibco], 1× Glutamax [Gibco], and 50 U/ml penicillin/streptomycin [Gibco]). Final cell suspension was obtained through mechanical dissociation with a P1000 pipette. Neurons were seeded and maintained at 37°C and 5% CO$_2$ in Neuronal Media (Neurobasal Medium-A [Gibco], supplemented with 1% B27 [Gibco], 1% Glutamax [Gibco], and 50 U/ml penicillin/streptomycin [Gibco]).

For astrocytes and microglia, cortexes were enzymatically digested with 10× Trypsin-EDTA without phenol-red (0.5%; Gibco) in HBSS for 13 min at 37°C. The enzymatic reaction was blocked adding equal volume of DMEM complete (Dulbecco's Modified Eagle's Medium [Gibco] supplemented with 10% FBS [FACS], 1% Glutamax [Gibco], 1% non-essential amino acids [Sigma-Aldrich], and 100 U/ml penicillin/streptomycin [Gibco]). The tissue suspension was pelleted at 232 × $g$ for 6 min, and the supernatant was removed. Final cell suspension was obtained through mechanical dissociation with a P1000 pipette and seeded into a T75 flask maintained at 37°C and 5% CO$_2$ in DMEM complete. After 4 d, the media was changed using DMEM complete.

To obtain microglia from this mixed culture, at 7 DIV, microglia proliferation was induced by adding 10 ng/ml GM-CSF (130-095-746; Miltenyi Biotec) to the DMEM complete. Microglia were harvested at 10-13 DIV by manually shaking flasks for 6 min and seeded on glass bottom dishes in DMEM without FBS (Dulbecco's Modified Eagle's Medium [Gibco] supplemented with 1% Glutamax [Gibco], 1% non-essential amino acids [Sigma-Aldrich], and 0.5% penicillin/streptomycin [Gibco]). Astrocytes were detached form the same flask using Trypsin-EDTA (0.05%; Gibco) for 10 min at 37°C.

### Cell lines

HEK293 and RAW264.1 cell lines were cultured in DMEM supplemented with glutamine. A549 cells were cultured in RPMI

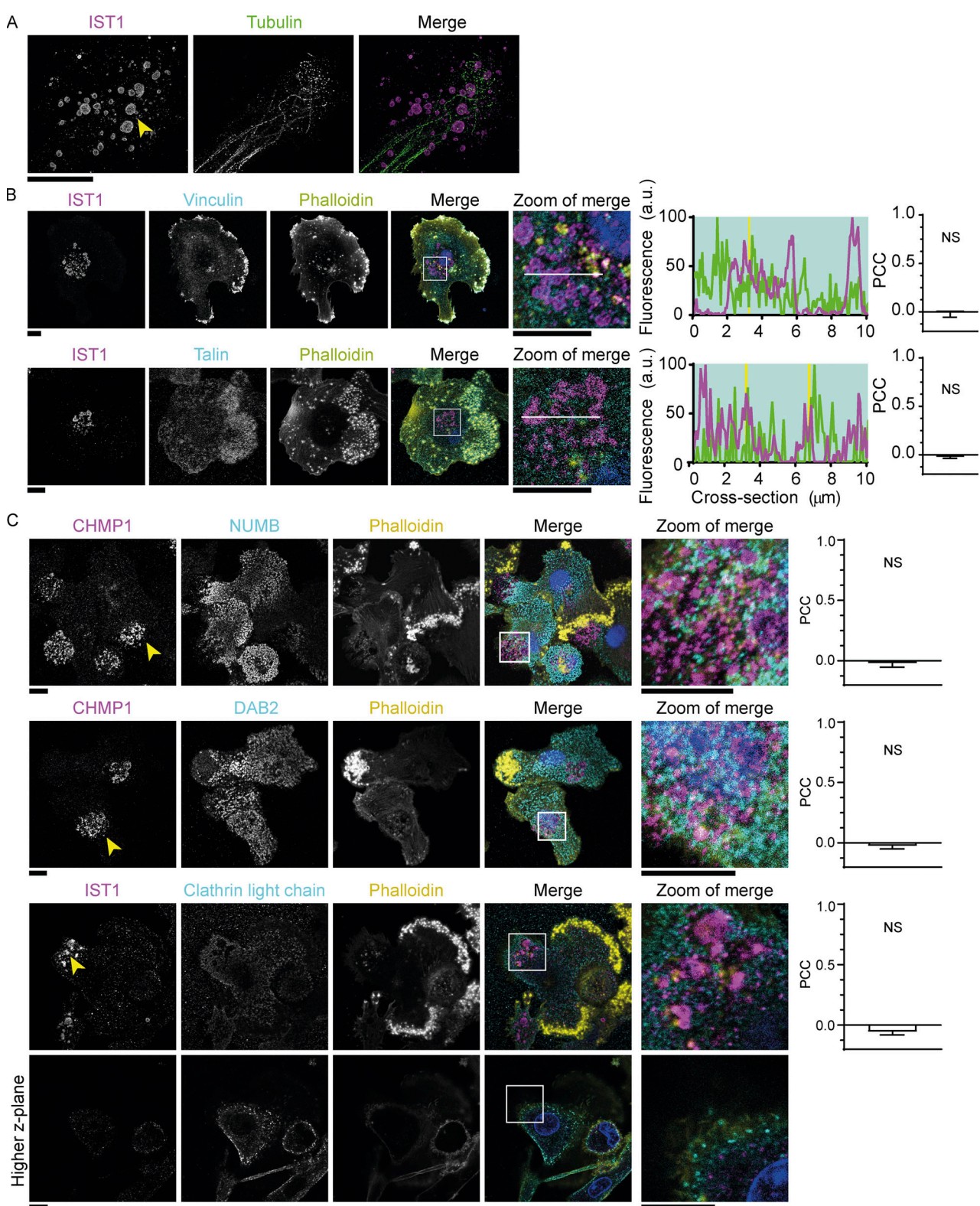

Figure 9. **IST1 structures are not reticular adhesions. (A)** STED micrographs of moDCs immunolabeled for IST1 (magenta) and tubulin (green). Scale bar: 10 µm. **(B)** Confocal micrographs of moDCs immunolabeled for IST1 (magenta), vinculin or talin (cyan), and phalloidin (yellow). Blue: DAPI. Bar graphs show PCC ± SD (n = 3 donors). PCC values were compared to 0 using one sample *t* test. **(C)** Confocal micrographs of moDCs immunolabeled for CHMP1 (magenta) and the reticular adhesion markers NUMB, DAB2 or clathrin light chain (cyan) and phalloidin (yellow), and DAPI (blue). For clathrin light chain, the image showing a higher z-plane indicates that clathrin labeling was successful. PCC values were compared to 0 using one sample *t* test (n ≥ 3 donors). Scale bars: 10 µm. For the statistical analysis of B and C, data distribution was assumed to be normal, but this was not formally tested. Scale bars: 10 µm.

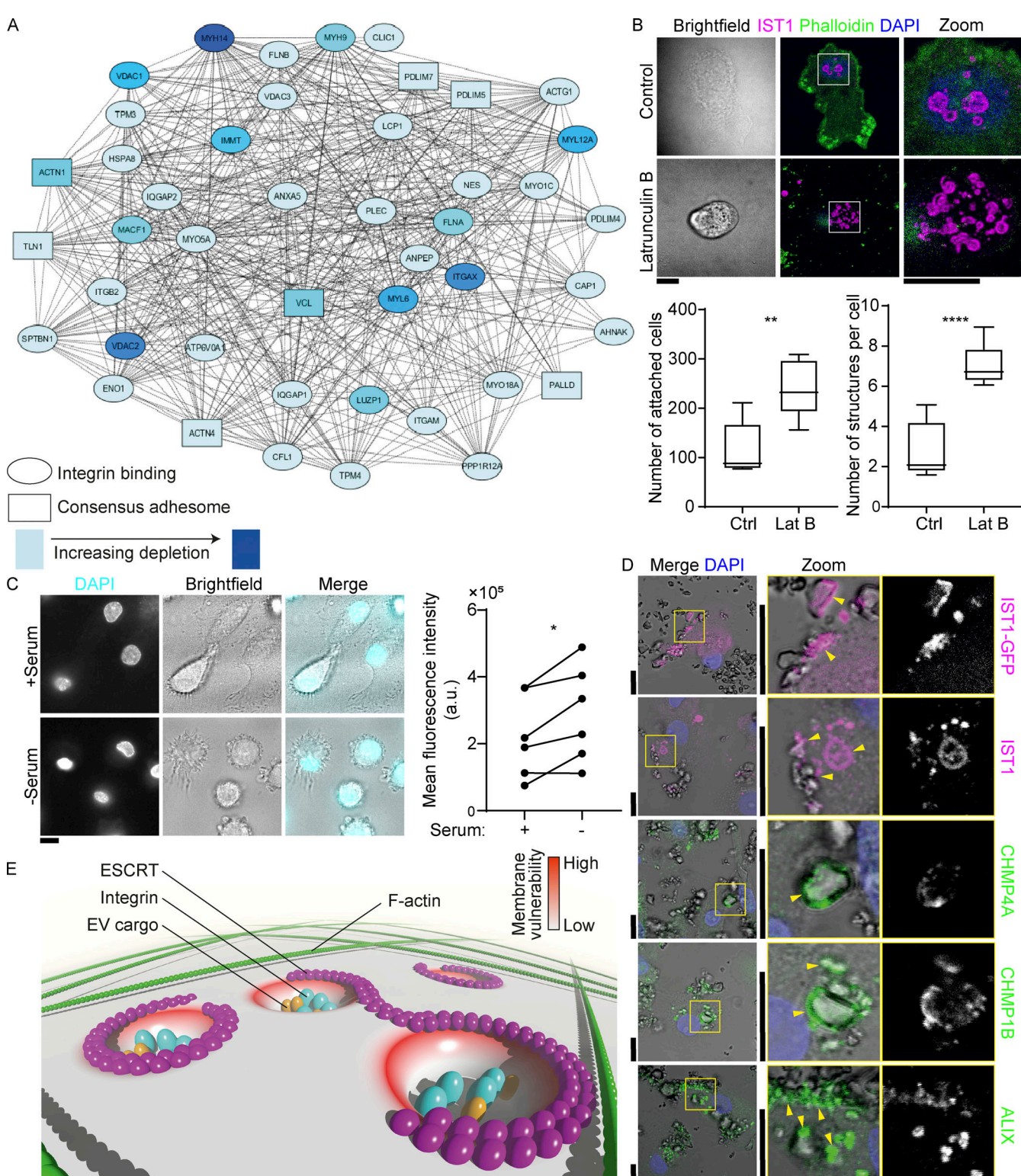

**Figure 10.** **IST1 structures are recruited to sites of membrane damage. (A)** STRING protein network of consensus adhesion proteins (Horton et al., 2015) identified by DAVID gene ontology that were depleted in response to 100 µM latrunculin B (Lat B) treatment. **(B)** Confocal micrographs of moDCs pre-incubated with or without Lat B. After washing away the Lat B containing medium, cells were seeded and incubated for 1 h followed by gentle washing with PBS to remove non-adherent cells. Left graph: average number of adhering moDCs. Right graph: average number of IST1 structures per cell. (n = 5 donors; >80 cells per donors; paired 2-sided t test). **(C)** Confocal micrographs of moDCs cultured with and without serum and in the presence of 0.5 µg/ml DAPI. Graph shows fluorescent intensities of DAPI. Data points show individual donors (>50 cells per condition; paired two-sided t test; *: P < 0.05; **: P < 0.01; ****: P < 0.0001; NS: not significant). **(D)** Confocal micrographs showing recruitment of ESCRT structures to plasma membrane contact sites (arrow heads) with membrane-disrupting silica crystals. **(E)** Model scheme with proposed mechanism. Integrins and other known extracellular cargo proteins (CD63, GPI-anchored proteins, ubiquitinated proteins) are enriched in membrane domains surrounded by ESCRT structures. The cortical F-actin cytoskeleton is disassembled at

these clusters. The integrin clusters tightly adhere to the extracellular substrate, making the surrounding membrane vulnerable to damage. This results in formation of the ESCRT structures. ESCRT repairs the membrane by shedding of damaged plasma membrane regions. For the statistical analysis of B and C, data distribution was assumed to be normal, but this was not formally tested. Scale bars: 10 µm.

1640 containing glutamine. HeLa cells were maintained in high-glucose DMEM with Glutamax (31966021; Gibco). HeLa cells were subconfluent at the time of fixation. All media was supplemented with AA and 10% FBS except in experiments where FBS was not desired with regard to the formation of ESCRT structures.

#### Immunofluorescence microscopy
For the microscopy experiments, 50,000 cells were plated on 12-mm-diameter glass coverslips in RPMI 1640 medium supplemented with L-glutamine without FBS unless stated otherwise. PBLs, HEK293 cells, A549 cells, and neurons were cultured on coverslips coated with poly-L-lysine (PLL; P4707; Sigma-

Aldrich) for 15 min. Cells were incubated at 37°C and 5% $CO_2$ for 2 h unless stated otherwise. Cells were fixed in 4% paraformaldehyde (PFA) for 15 min at RT or in ice cold methanol at −20°C for 5 min. Cells were blocked and permeabilized in PBS with 20 mM glycine (Bachem), 3% (w/v) BSA (9048-46-8; Fisher Scientific), and 0.1% (w/v) saponin (47036-50G-F; Sigma-Aldrich) for 30 min at RT. Cells were incubated with primary antibody in this blocking solution overnight at 4°C (Table 2). Subsequently, cells were washed in PBS three times, and coverslips were incubated for 1 h at RT with a combination of the following secondary antibodies: donkey anti-mouse IgG (H+L) Alexa Fluor 647 (A31571; Thermo Fisher Scientific, Invitrogen), donkey anti-rabbit IgG (H+L) Alexa Fluor 488 (A21206; Thermo

Table 1. **Triple fluorescence staining panel used for immunohistochemistry**

| | Targets | | | |
|---|---|---|---|---|
| | Vimentin | IST1 | CD68 | Nucleus |
| Isotype (primary ab) | Rabbit polyclonal (ab92547; Abcam) (1:250 dilution) | Mouse IgG2a (66989-1-Ig; Proteintech) (1:750 dilution) | Mouse IgG3 (M0876; DAKO) (1:100 dilution) | |
| Secondary ab | Donkey anti-rabbit IgG (ab150075; Abcam) (1:50 dilution) | Rat anti-mouse IgG2a (RMG2a-62; BioLegend) (1:20 dilution) | Goat anti-mouse IgG3 (1101-01; Southern bio) (1:75 dilution) | |
| Tertiary ab | | Donkey anti-rat IgG (ab150153; Abcam) (1:40 dilution) | Donkey anti-goat IgG (ab175704; Abcam) (1:75 dilution) | |
| Conjugate/dye | AF647 | AF488 | AF568 | DAPI |
| Longpass filter | 610LP | 515LP | 590LP | 420LP |
| | Targets | | | |
| | CD18 | IST1 | CD68 | Nucleus |
| Isotype (primary ab) | Rabbit polyclonal (a2173; Abclonal) (1:150 dilution) | Mouse IgG2a (66989-1-Ig; Proteintech) (1:750 dilution) | Mouse IgG3 (M0876; DAKO) (1:100 dilution) | |
| Secondary ab | Donkey anti-rabbit IgG (ab150075; Abcam) (1:50 dilution) | Rat anti-mouse IgG2a (RMG2a-62; BioLegend) (1:20 dilution) | Goat anti-mouse IgG3 (1101-01; Southern bio) (1:100 dilution) | |
| Tertiary ab | | Donkey anti-rat IgG (ab150153; Abcam) (1:40 dilution) | Donkey anti-goat IgG (ab175704; Abcam) (1:100 dilution) | |
| Conjugate/dye | AF647 | AF488 | AF568 | DAPI |
| Longpass filter | 610LP | 515LP | 590LP | 420LP |
| | Targets | | | |
| | CD18 | IST1 | Vimentin | Nucleus |
| Isotype (primary ab) | Rabbit polyclonal (a2173; Abclonal) (1:150 dilution) | Mouse IgG2a (66989-1-Ig; Proteintech) (1:750 dilution) | Mouse IgG1 (RV202; Santa Cruz) (1:300 dilution) | |
| Secondary ab | Donkey anti-rabbit IgG (ab150075; Abcam) (1:50 dilution) | Rat anti-mouse IgG2a (RMG2a-62; BioLegend) (1:20 dilution) | Goat anti-mouse IgG1 (1071-01; Southern bio) (1:100 dilution) | |
| Tertiary ab | | Donkey anti-rat IgG (ab150153; Abcam) (1:40 dilution) | Donkey anti-goat IgG (ab175704; Abcam) (1:100 dilution) | |
| Conjugate/dye | AF647 | AF488 | AF568 | DAPI |
| Longpass filter | 610LP | 515LP | 590LP | 420LP |

Fisher Scientific, Invitrogen), donkey anti-mouse IgG (H+L) Alexa Fluor 488 (A21202; Thermo Fisher Scientific, Invitrogen), goat anti-rat IgG (H+L) Alexa Fluor 488 (A11006; Thermo Fisher Scientific, Invitrogen), donkey anti-rabbit IgG (H+L) Alexa Fluor 647 (A31573; Thermo Fisher Scientific, Invitrogen), and donkey anti-mouse IgG (H+L) Alexa Fluor 647 (A31571; Thermo Fisher Scientific, Invitrogen), all at 1:800 dilution. For actin labeling, phalloidin Alexa Fluor 546 was used (A22283; Thermo Fisher Scientific) at 1:200 dilution in parallel with the labeling with secondary antibodies. Cells were mounted in 70% glycerol supplemented with DAPI. Imaging was performed on a LSM800 Zeiss confocal laser scanning microscope with a 20× air or 63 × 1.4 NA oil immersion objective, unless stated otherwise. Images were analyzed using FIJI-ImageJ (Schindelin et al., 2012).

### Immunofluorescence microscopy of brain cells
All cells were seeded at a density of 200,000 cells per well on 35-mm glass bottom dishes with 20 mm bottom well and 1.5 thickness (Cellvis). 6 h upon seeding, hippocampal neurons, astrocytes, and microglia were fixed using ice cold 4% paraformaldehyde for 15 min at 4°C. Permeabilization and blocking was carried out as described above but at 4°C. Cells were incubated in anti-IST1 IgG at a concentration of 1:500 (A305-411A; Thermo Fisher Scientific, Bethyl Laboratories) overnight at 4°C and labeled with secondary antibody and mounted as described above. Laser-scanning confocal microscopy was carried out on a Nikon Scanning Confocal A1Rsi+ inverted laser microscope equipped with Transmission light detector (DIC, phase), Spectral scans, Point scanner (Galvo), two detectors Hybrid GaAsP PMTs (high sensitivity, GFP, mCherry), and an objective PL APO 60×/1.4NA oil immersion lens. Images were collected as Z-series with thickness of 0.3 µm at a resolution of 1,024 × 1,024. For each image, three laser channels with wavelength intensities of 405- (DAPI), 488- (Alexa Fluor 488), and 561- (Alexa Fluor 568) nm, respectively, were collected. The bright field image was composed using the transmission light detector (DIC). Images were acquired using Acquisition software NIS Elements AR 5.21.02 and then analyzed with FIJI-ImageJ (Schindelin et al., 2012).

### RNA labeling
MoDCs were cultured on coverslips in RPMI lacking FBS for 1 h. Subsequently, they were fixed in methanol for 10 min at –20°C and washed three times with PBS. Next, cells were labeled in 500 nM SYTO RNA Select green fluorescent cell stain solution (S32703; Thermo Fisher Scientific) in PBS for 20 min at RT. Subsequently, cells were three times washed with PBS and blocked and immunolabeled as described above.

### Memglow 640 labeling
MoDCs were fixed in 4% paraformaldehyde for 15 min at RT. After three washes with PBS, the cells were permeabilized by incubation with 0.1% Triton X-100 in PBS for 5 min at RT. Cells were immunolabeled as described above. Next, cells were incubated in freshly made 100 nM Memglow 640 solution in PBS (Cytoskeleton, MG04) for 10 min at RT. Cells were washed with PBS twice and mounted as described above.

### 3D cell culturing and immunofluorescence labeling
To prepare 600 µl of collagen matrix, 400 µl cold rat tail collagen I (A1048301; Thermo Fisher Scientific, Gibco) was added to a mix of 125 µl water, 15 µl 1 M NaOH and 60 µl 10× PBS on ice. Subsequently, 60 µl of collagen mix was pipetted to a glass bottom dish (Will-Co wells, HBST-3512). For the experiments with labeled collagen, 1:100 FITC-labeled collagen (C4361; Sigma-Aldrich) was added. The dish was kept at RT for 1 min before incubating at 37°C and 5% $CO_2$ for 1 h to allow the collagen to polymerize. Then, 2 ml of cell suspension (50,000 cells/ml in RPMI supplemented with 10% FBS) was added on top of each collagen matrix. The cells were incubated at 37°C and 5% $CO_2$ for 5 h. Subsequently, they were fixed using 4% PFA for 15 min at RT. The matrices were washed with PBS followed by permeabilization and blocking in PBS with 20 mM glycine, 3% (w/v) BSA, and 0.1% (w/v) saponin for 1 h at RT. Subsequently, the samples were immunolabeled as described above. Mounting medium (see above) containing DAPI was applied on top of the matrices, and the collagen was protected from drying out by placement of a coverslip on top of the matrix. Imaging was performed using a LSM800 Zeiss confocal laser scanning microscope with a 63 × 1.4 NA oil immersion objective. Images were analyzed using FIJI-ImageJ (Schindelin et al., 2012).

### Silica particles
Silica particles were from USA Silica (Min-U-Sil10) and incubated at ~10 particles per cell for 1 h followed by fixation and immunofluorescence labeling.

### STED imaging
Samples were prepared as described above for confocal laser scanning microscopy, but with use of a combination of the following secondary antibodies: goat anti-rabbit IgG Abberior STAR 580 (41367; Sigma-Aldrich), goat anti-rabbit IgG Abberior STAR 635 (41348; Sigma-Aldrich), goat anti-rat Abberior STAR 580 (2-0132-005-1, #03092018Hp; Abberior), and goat anti-mouse Abberior STAR 580 (ST580-1001-500UG, #90923CW-8; Abberior).

Images were acquired using an Abberior Expert Line microscope with a 100× oil immersion objective (Olympus Objective UPlanSApo 100×/1.40 Oil) and 20°C refractive index matching oil for high-resolution imaging. Prior to imaging a 0.1 µm TetraSpeck (T7279; Invitrogen) bead sample was used to align confocal lasers and optimize the STED doughnut position.

All 2D STED images were acquired with a pixel size of 20 nm and a pinhole size of 0.9 A.U. For excitation and depletion, we used a 40 MHz pulsed laser, with 750 ps delay and a gated detection window of 8 ns. The used lasers had powers of 200 µW (561 nm), 1 mW (640 nm), and >2,750 mW (775 nm STED) out of the laser heads. For the confocal images, 10 µs of 0.2% 640 laser intensity was used, and 10 µs of 30% laser intensity of the 561 laser. For the STED images, 1% of 640 laser intensity and 65% of STED775 laser intensity with a dwell time of 100 µs was used. For the second STED color, 30% of 561 laser intensity and 80% of STED775 laser intensity with a dwell time of 100 µs was used. For the 640 laser excitation, the corresponding avalanche

Table 2. **Primary antibodies used for samples for confocal imaging**

| Antibody | Host | Dilution for microscopy | Supplier | Catalog nr. | Notes |
|---|---|---|---|---|---|
| ALG-2 | Rabbit | 1:200 | Proteintech | 12303-1-AP | |
| ALIX | Mouse | 1:100 | BioLegend | 634502 | |
| CABIN1 | Rabbit | 1:100 | NovusBio | NBP1-91745 | |
| CD63 | Mouse | 1:200 | Santa Cruz | sc-59284 | |
| CD9 | Rabbit | 1:100 | Abcam | ab97999 | |
| Calprotectin heterodimer | Rabbit | 1:100 | R&D systems | MAB45701-SP | |
| CHMP1B | Mouse | 1:50 | Santa Cruz | sc-514013 | Used for CLSM, STED, and TEM |
| CHMP4A | Rabbit | 1:100 | Thermo Fisher Scientific | PA5-49410 | |
| Clathrin light chain | Mouse | 1:500 | Sigma-Aldrich | C1985 | |
| DAB2 | Rabbit | 1:100 | CST | 129065 | |
| EEA1 | Mouse | 1:100 | BD Biosciences | 610456/610457 | |
| ERGIC53 | Mouse | 1:200 | Enzo Life Sciences | enz-ABS300-0100 | |
| GFP | Rabbit | 1:200 | Abcam | ab6556 | |
| GM130 | Mouse | 1:200 | BD | 610822 | |
| Histone H3 | Rabbit | 1:200 | Abcam | ab5103 | Methanol fixation |
| HLA-DP/DQ/DR | Mouse | 1:100 | NSJ Bio | V3374 | |
| HLA-DR | Mouse | 1:100 | BioLegend | 307614 | |
| HLA-ABC | Rabbit | 1:200 | Thermo Fisher Scientific | 14-9983-82 | |
| HNRNPC | Rabbit | 1:200 | Abcam | ab133607 | |
| Integrin β1 | Rat | 1:200 | Gift of Carl Figdor | Clone: AIIB2 | |
| Integrin β2 | Mouse | 1:200 | Gift of Carl Figdor | Clone: L19 | |
| Integrin αVβ5 | Mouse | 1:200 | Merck Millipore | Clone 15F11 | |
| Integrin αM | Rat | 1:200 | BioLegend | 101204 | |
| IST1 | Mouse | 1:100 | Proteintech | 66989-1-Ig | Methanol fixation |
| IST1 | Rabbit | 1:1,000 | Bethyl | A305-411A | |
| Lamin A/C | Rabbit | 1:200 | Abcam | ab108595 | |
| LAMP1 | Mouse | 1:200 | BioLegend | 328601 | |
| LC3 | Rabbit | 1:100 | CST | 2775S | |
| NUMB | Rabbit | 1:100 | CST | 2756S | |
| PDI | Rabbit | 1:100 | Novus Bio | NB300-517 | |
| S100a8 | Mouse | 1:200 | Proteintech | 66853-1-Ig | |
| S100a9 | Rabbit | 1:200 | Thermo Fisher Scientific | PA1-46489 | |
| Sec22b | Rabbit | 1:100 | Novus Biologicals | NB100-91277 | |
| Spastin | Rabbit | 1:100 | Novus Biological | NBP1-90230 | |
| Talin | Mouse | 1:100 | Sigma-Aldrich | T3287 | |
| TAPBP | Rabbit | 1:100 | LS Bio | LS-C331792 | |
| TOMM20 | Mouse | 1:100 | Abcam | ab56783 | |
| TSG101 | Mouse | 1:50 | BD | BD61296 | |
| Tubulin | Rat | 1:500 | Novus Biologicals | NB100-1639 | Used for CLSM and STED |
| Ubiquitin | Mouse | 1:200 | Enzo Life Sciences | BML-PW8810-100 | |
| Vinculin | Mouse | 1:300 | Sigma-Aldrich | V9131 | |
| VPS4 | Rabbit | 1:100 | Abcam | ab180581 | In some cases this labeled the cytoskeleton. |

photodetector was set to 650–763 nm and for 561 laser excitation to 570–630 nm.

For the 3D STED images, the pixel size was set to 35 nm in the xy plane and 35 nm in z and 0.7 A.U., and the STED775 laser was put at 100% 3D mode. For excitation and depletion, we used a 40 MHz pulsed laser with 750 ps delay and a gated detection window of 15 ns. For the confocal overview, 0.1% of 640 laser intensity was used, and for 3D STED 0.8% of 640 laser intensity and 95% of STED775 laser intensity was used with a dwell time of 150 µs. The detectors were set to detect between 650 and 730 nm. Additionally, we used the adaptive illumination technique, RESCue (Reduction of State transition Cycles), to reduce photobleaching (Staudt et al., 2011). With this technique, intensity thresholds can be set at specific durations of the dwell time; if the threshold (number of counts) is not surpassed, the laser is switched off for the remainder of the dwell time.

## Correlative light and electron microscopy

Cells cultivated in glass bottom µ dishes (81148; Ibidi) were fixed with a mixture of 2% formaldehyde and 0.2% glutaraldehyde in 0.1 M pH 7.4 phosphate buffer. Cells were blocked and permeabilized with PBS containing 3% BSA, 10 mM glycine, and 0.1% saponin. Cells were incubated with antibodies against CHMP1B (sc-514013; Santa Cruz) overnight at 4°C, washed three times with 0.1 M pH 7.4 PBS, and labeled for 2 h at RT with FluoroNanogold (Nanoprobes-7502). Cells were imaged with a LSM800 Zeiss confocal laser scanning microscope Airy scan equipped with a 32-channel gallium arsenide phosphide photomultiplier tube (GaAsP-PMT), Zen 2009 software (Carl Zeiss), and a 63 × 1.40 NA objective (Carl Zeiss). Fluorescence of FluoroNanogold was visualized by excitation with a 640 nm laser.

Silver enhancement was performed for intensifying the gold particles (Aurion-500.033). Cells were post fixed with 2% glutaraldehyde for 1 h at RT and 1% OsO4 for 30 min at RT in 0.1 M pH 7.2 sodium cacodylate buffer. Samples were en bloc stained with 0.5% uranyl acetate and dehydrated in graded series of ethanol. Samples were flat embedded in epon.

After removal of the µ dish, 100-nm serial sections of the region of interest were collected on formvar-coated and carbon-evaporated copper single slot grids. Serial sections were contrasted with uranyl acetate and lead citrate and imaged with a CM12 transmission electron microscope (Philips) operating at 100 kV. EM images were aligned with the Airyscan images using the eC-CLEM plugin (Paul-Gilloteaux et al., 2017) in Icy (http://icy.bioimageanalysis.org). The grid containing the section with structures of interest was decorated with 10 nm gold particles (752584; Sigma-Aldrich) to enable alignment of the individual images to form the tomogram. Double-tilt tomography series including a tilt range of +40° to –40° with 2.5° increments were generated. The IMOD software package was used for reconstructing the tomograms.

## Transfections

MoDCs were transfected by electroporation using the NEON Transfection System (Invitrogen) with 100 µl NeonTips. The protocol entailed 2 pulses of 1,000 V and 40 ms.

### Overexpression

The concentration of DNA used was 3 µg per 1 million cells. The IST1-GFP construct was generated by cloning IST1 into the pEGFP-N1 vector (sequence can be found in supplemental materials, Data S1) and is available at Addgene (#186298). The CHMP4B-mCherry construct was a gift from Ana Jimenez Joaquina (Institut Curie, Paris, France) and has been described (Jimenez et al., 2014). The GPI-mRFP construct, coding for GPI-anchored monomeric RFP behind the CMV promoter (Nadler et al., 2013), was a gift from André Nadler (Max Planck Institute of Molecular Cell Biology and Genetics, Dresden, Germany). GFP-C1-PLCdelta-PH was a gift from Tobias Meyer (Cornell University, Ithaca, NY, USA; plasmid #21179; Addgene) and has been described (Stauffer et al., 1998). pcDNA3-AKT-PH-GFP has been described (Kwon et al., 2007). Integrin beta 2 - mYFP was a gift from Timothy Springer (Harvard Medical School, Boston, MA, USA; plasmid #8638; Addgene) and has also been described (Kim et al., 2003). NCF4-PX-EGFP was a gift from Michael Yaffe (Massachusetts Institute of Technology, Cambridge, MA, USA; plasmid #19010; Addgene; Kanai et al., 2001). Myc-ALIXΔPRR-mCherry was a gift from Aurélien Roux (University of Geneva, Geneva, Switzerland) and has been described (Larios et al., 2020). mCherry-caveolin-C-10 was a gift from Michael Davidson (Florida State University, Tallahassee, FL, USA; plasmid #55008, Addgene) and has been described (Hanson et al., 2013). We previously generated the MCOLN1-N construct (residues 1–68 of mouse MCOLN1; Baranov et al., 2016) and this construct is available on Addgene (#92419). After electroporation, the cells were cultured at ~300,000 cells per well in an ultra-low attachment plate (3473; Sigma-Aldrich, Costar) in phenol-red free RPMI containing 20% FBS for 4 h.

### Knockdown

The concentration of siRNA used was 3 µg siRNA per 1 million cells. Cells were transfected with non-targeted (NT) siRNA (12935300; Thermo Fisher Scientific), a mix of four siRNAs targeting IST1 (HSS114768, HSS190628), a mix of six siRNAs targeting TSG101 (HSS111013, HSS111014, and HSS186437), a mix of six siRNAs targeting HRS (HSS113536, HSS113537, and HSS113538), or a mix of six siRNAs targeting ALIX (HSS115204, HSS115205, and HSS115206). Cells were incubated in phenol red-free RPMI for 3 h in an ultra-low attachment plate at 37°C. Subsequently, phenol-red free RPMI containing a double amount of FBS and cytokines was added, resulting in a final concentration of 10% FBS, 300 U/µl IL-4, 450 U/µl GM-CSF, and 1% AA, and cells were incubated for 2 d at 37°C and 5% CO$_2$. Before reseeding in the absence of FBS, the cells were harvested by pipetting medium directly on the cells and the FBS was washed away.

### TIRF imaging

4 h after transfection, cells were harvested by pipetting medium directly on the cells and the FBS-containing medium was washed away. Subsequently, cells were seeded in RPMI lacking phenol red and FBS in a four-compartment glass-bottom dish (627870; Greiner). After ~30 min of incubation at 37°C and 5% CO$_2$, the cells were imaged at RT with a home-built TIRF microscope

using a 490 nm laser for excitation, an Olympus 60× UAPO NA 1.49 Oil objective, and a Prime BSI Express Scientific CMOS camera (Photometrics). The microscope was controlled with μManager (Edelstein et al., 2010), and images were analyzed using FIJI-ImageJ (Schindelin et al., 2012).

## Coating coverslips
Coverslips were incubated in 500 μl of FBS for 10 min at RT, poly-L-lysine for 15 min at RT, 1% rat tail collagen (A1048301; 3 mg/ml; Gibco) in PBS overnight at 4°C or 1% fibronectin bovine plasma (F1141; 1 mg/ml; Sigma-Aldrich) in PBS overnight at 4°C. Next, the coverslips were washed three times with PBS, and cells were seeded as described above.

## Inhibiting actin polymerization
MoDCs were pre-incubated in a tube with 100 μM Latrunculin B (ab144291; Abcam) in RPMI for 30 min at 37°C and 5% $CO_2$. As solvent control, an identical volume of ethanol was added to the control samples. After pre-incubation, Latrunculin B and ethanol were washed away. Next, the cells were seeded on glass coverslips in a 24-well plate at 70,000 cells per well and incubated for 1 h at 37°C.

## Adherence assay
After incubation, coverslips were gently washed with PBS to remove non-adherent cells. Next, cells were fixed with 4% PFA and immunolabeled for IST1, phalloidin, and DAPI as described above. Using an LSM800 Zeiss confocal laser scanning microscope with a 20× air objective, a set area was imaged for all conditions, and the number of cells in this area was counted manually.

For the EDTA experiment, cells were seeded on poly-L-lysine-coated coverslips and incubated for 1 h with or without 5 mM EDTA.

## Western blotting
For the quantification of knockdowns using Western blot, 500,000 cells treated with siRNA were washed using PBS twice and subsequently lysed in 1% SDS, 10 mM Tris-HCl, pH 6.8, with protease inhibitors (11697498001; Roche). Lysates were run on 4–20% Mini-Protean TGX precast gels (4561094; Bio-rad) and blotted to PVDF membranes. Blots were incubated in Ponceau S (P3504; Sigma-Aldrich), washed in demiwater, imaged, and analyzed with Fiji (ImageJ). After blocking, blots were probed with polyclonal rabbit anti-IST1 IgG (A305-411A; Bethyl) at 1:20,000, monoclonal mouse anti-TSG101 IgG (6129690; BD Biosciences) at 1:500, or monoclonal mouse anti-ALIX IgG1 (634502; BioLegend) at 1:1,000. Subsequently, blots were labeled with goat anti-rabbit IgG IR-Dye800CW (926-32211; LI-COR) and donkey anti-mouse IgG IR-Dye680CW (926-32222; LI-COR), both at 1:5,000. Blots were scanned using the Odyssey CLx Infrared Imaging System and analyzed using ImageStudio (both LI-COR). Blots from the immunoprecipitation of IST1 (see below) were also probed for integrin β2 (polyclonal goat IgG anti-integrin beta 2/CD18; AF1730; R&D Systems) at 1:1,000.

## Stimulation using lipopolysaccharide
MoDCs, neutrophils, and monocytes were cultured in the presence or absence of 100 ng/ml lipopolysaccharide (LPS; L4391; Sigma-Aldrich) for 4 h.

## DAPI leakage assay
For the DAPI leakage experiment, 100,000 cells resuspended in phenol-red free RPMI medium were seeded in a Willco-dish. For each donor, there was a condition with 10% FBS and without FBS. Next, DAPI was added at a concentration of 0.5 μg/ml, after which the cells were incubated for 2 h at 37°C. After incubation, the medium was replaced by phenol-red free RPMI with or without FBS. The samples were imaged with a Zeiss Axio Observer Z1 and analyzed by measuring the fluorescence intensities of the nucleus in the DAPI channel by manually selecting the nucleus in Fiji (ImageJ).

## Flow cytometry
Cells were first labeled with polyclonal goat IgG anti-integrin beta 2/CD18 (AF1730; R&D Systems), and subsequently with FITC-labeled anti-human CD11b (301330; BioLegend) and donkey anti-goat IgG Alexa Flour 647 (A21447; Thermo Fisher Scientific). Data were acquired on a Cytoflex LX flow cytometer (Beckman Coulter) and analyzed using FlowJo (BD Biosciences).

## Immunoprecipitation and mass spectrometry
MoDCs were incubated at 37°C and 5% CO2 overnight in RPMI 1640 with glutamine and AA, supplemented with or without 10% FBS. Two million cells per condition were seeded. The next day, the cells were lysed using lysis buffer (200 mM NaCl, 75 mM Tris-HCl, 15 mM NaF, 7.5 mM EDTA, 7.5 mM EGTA, 0.15% Tween20). The lysates were pipetted up and down using a 25 g micro lance (300600; BD) at least two times to disrupt free DNA. The lysate was pelleted at >10,000 g for 10 min. Next, the lysates were incubated with rabbit IST1 antibody (A305-411A; Bethyl) for 30 min at 4°C at a concentration of 7.5 μg antibody per million lysed cells. Subsequently, 136.3 μg washed protein A/G magnetic beads (88802; Pierce) were added per 7.5 μg IST1 antibody (corresponding to a concentration of 1 mg beads per 55 μg antibody). The mix was incubated for 2 h at 4°C. Next, the lysis buffer was removed, and the beads were washed. The beads were incubated two times with elution buffer (0.2 M glycine, pH between 2 and 3) for 10 min at RT. The lysate was quenched using pH 8 Tris-HCl solution. Protein mixtures were briefly run on a Bis-Tris polyacrylamide gel. The gel was then briefly stained in Instant Blue dye (Expedion) and washed in milliQ distilled water. After incubation in milliQ water overnight, the stained protein-containing fraction of the acrylamide gel was excised using sterile blades. Subsequently, each sample was sliced into six evenly sized pieces and transferred to a well of a perforated plate. To remove the Instant Blue dye, the gel was washed twice in a 1:1 mixture of 100% acetonitrile and 25 mM $NH_4HCO_3$ solution. The samples were then subject to Trypsin-based digestion (Humphries et al., 2009). Next, the samples were de-salted prior to mass-spectrometry-based analysis using R3 POROS beads. The peptides were then resuspended in a 5% (v/v) acetonitrile 0.1% (v/v) formic acid solution.

### Data analysis methodology
The raw data files from the instrument were processed using Thermo Fisher Scientific's Proteome Discoverer (PD) software (version 2.3.0.532). The data were processed using the consensus workflow provided with PD in the file CWF_Comprehensive_Enhanced Annotation_LFQ_and_Precursor_Quan and the

processing workflow provided with PD in the file PWF_QE_-Precursor_Quan_and_LFQ_SequestHT_Percolator. The processing workflow was set to search 350,097 spectra against the protein database *Homo sapiens* (SwissProt TaxID = 9606; v2017-10-25) and *Homo sapiens* (TrEMBL TaxID = 9606; v2017-10-25) using the Sequest HT (version that was provided with the version of PD) software. The protein identification search engine parameters used were:

- the enzyme Trypsin (cleaving at Lysines and Arginines except where the presence of a C-terminal Proline obstructs cleavage)
- the precursor tolerance of 20 ppm and a fragmentation tolerance of 0.5 Da
- the fixed modifications of Carbamidomethyl (+57.021 Da) to Cysteine and variable modifications of Oxidation (+15.995 Da) to Methionine.

A false discovery rate (FDR) is calculated for both the protein level and the peptide level by PD. Proteins were labeled with high confidence where the FDR is < 0.01; Medium confidence where the FDR is between 0.01 and 0.05; and low confidence where the FDR is > 0.05. A target FDR cut-off of 0.01 was applied.

## Mass spectrometry for latrunculin B treatment

Peptide solutions were acidified by adding 0.1% TFA and purified, prior to mass spectrometry, by STAGE tips (Rappsilber et al., 2003). For mass spectrometry, purified peptides were separated on a Dionex Ultimate 3000 RSLC nano flow system (Dionex). A 3 µl of sample was loaded in 0.1% trifluoroacetic acid (TFA) and acetonitrile (2% acetonitrile in 0.1% TFA) onto an Acclaim Pep Map 100 µm × 2 cm, 3 µm C18 nano trap column, at a flow rate of 5 µl/min, bypassing the analytical column. Elution of bound peptides was performed with the trap column inline with an Acclaim PepMap C18 nano column 75 µm × 25 cm, 3 µm, 100 Å (Analytical Column) with a linear gradient of 96% buffer A and 4% buffer B to 60% buffer A and 40% buffer B, (buffer A: 0.5% Acetic Acid; buffer B: 80% acetonitrile in 0.5% acetic acid) at a constant flow rate of 300 nl/min over 60 min. The sample was ionized in positive ion mode using a Proxeon nano spray ESI source (Thermo Fisher Scientific) and analyzed in an Orbitrap Velos Pro FTMS (Thermo Finnigan). The Orbitrap Velos Pro instrument under Xcalibur2.1 software was operated in the data-dependent mode to automatically switch between MS and MS/MS acquisition. MS spectra of intact peptides (m/z 350-1,600) with an automated gain control accumulation target value of 1,000,000 ions were acquired with a resolution of 60,000. The 10 most abundant ions were sequentially isolated and fragmented in the C trap, where dissociation was induced by HCD mode, using an accumulation target value of 10,000 and a normalized collision energy of 45%. Dynamic exclusion of ions sequenced within the 45 previous seconds was applied. Unassigned charge states and singly charged ions were excluded from sequencing. The ion selection threshold was 10,000 counts for MS2.

### Data analysis by Max Quant

Peptides and proteins were identified by Andromeda via automated database searching of all tandem mass spectra against a curated target/decoy database (forward and reversed version of the Human protein sequence database (http://www.uniprot.org/, UniProt. Release January 2022) containing all Human protein entries from Swiss-Prot and TrEMBL). Spectra were initially searched with a mass tolerance of 6 ppm in MS mode and 0.5 Da in MS/MS mode and strict trypsin specificity and allowing up to 2 missed cleavage sites. Cysteine carbamidomethylation was searched as a fixed modification, whereas N-acetyl protein, deamidated NQ, oxidized methionine were searched as variable modification. The resulting Andromeda peak list-output files were automatically loaded into inbuilt MaxQuant software modules for further processing, label-free quantitation (LFQ) and a maximum FDR of 1% was fixed for the result output files. Downstream analysis of datasets was carried out by Perseus software suite version 1.6.13 downloaded from maxquant.org.

## Harvesting cells with cold procedure

MoDCs were harvested by incubating them in cold PBS at 4°C for 1 h and subsequently vigorously tapping the flask to detach the cells. When seeded on coverslips, cells were detached by pipetting PBS directly onto the cells instead of by tapping.

## Analysis
### Pearson correlation coefficient (PCC) analysis

In Fiji (ImageJ), the imaged cell areas were selected based on the phalloidin signal. Using the option Coloc 2, the PCC for the two different channels was calculated.

### Analysis of number and size of IST1 structures

For the analysis of the number and size of the ESCRT structures, microscope images were analyzed in Fiji (ImageJ) using a macro (Data S1). With this macro, structures larger than 0.3 µm² were automatically identified based on a fluorescent threshold. The number of cells containing the IST1-positive structures was determined by manual counting.

### Statistical methods

Data were analyzed with paired two-tailed Student's *t* tests (for comparisons of two conditions; pairing for donors), one-sample two-tailed *t* tests (for determining whether Pearson correlation coefficients differ from 0), or ANOVA followed by Dunnet's post hoc test (for testing >2 conditions), as defined in the figure legends. In all cases, data distribution was assumed to be normal, but this was not formally tested.

## Online supplemental material

Fig. S1 contains extra data showing that IST1 structures are located at the plasma membrane of cells cultured in collagen matrices and super-resolution microscopy of the structures. Fig. S2 contains extra data showing that LPS does not influence the IST1 structures and that they are not present in non-migratory non-adhering cell types. Fig. S3 contains extra data showing localization of IST1 structures with respect to organelle markers. Fig. S4 contains extra data showing that the structures surround GPI-anchored proteins, but do not colocalize with caveolin-1 and the ALIXΔPRR mutant. Fig. S5 shows extra data for IST1-

immunoprecipitation and proteomics. Fig. S6 contains extra data showing the presence of S100A8 at the IST1 structures. Fig. S7 shows latruculin B control experiments and knockdowns of ESCRT proteins. Video 1 shows three-dimensional confocal microscopy reconstructions of moDC cultured in 3D collagen matrix showing IST1 structures at the plasma membrane. Video 2 shows three-dimensional confocal microscopy reconstructions of moDC cultured in 3D collagen matrix showing IST1 structures at the plasma membrane. Video 3 shows three-dimensional confocal microscopy reconstructions of moDC cultured in 3D collagen matrix showing IST1 structures at the plasma membrane. Video 4 shows three-dimensional confocal microscopy reconstructions of moDC cultured in 3D collagen matrix showing IST1 structures at the plasma membrane. Video 5 shows time-lapse TIRF microscopy showing a moDC transfected with IST1-GFP with immobile IST1-positive structures. Video 6 shows time-lapse TIRF microscopy showing a moDC transfected with IST1-GFP with immobile IST1-positive structures. Video 7 shows time-lapse TIRF microscopy of a monocyte-derived dendritic cell (moDC) transfected with IST1-GFP recorded with high laser power, showing that the IST1-positive structures persist upon cell death. Table S1 shows IST1 pulldown mass spectrometry results. Data S1 shows the sequence of the IST1-GFP construct and of the Fiji macro for the analysis of the number and size of the ESCRT structures.

### Data availability

The data that support the findings of this study are deposited at Zenodo under DOI numbers: 10.5281/zenodo.7598239, 10.5281/zenodo.7598723, 10.5281/zenodo.7599728, 10.5281/zenodo.7600202, 10.5281/zenodo.7600209, and 10.5281/zenodo.7600213. The IST1-GFP construct is available on Addgene (plasmid #186298).

## Acknowledgments

We thank Dr. Arjan Diepstra from the department of Pathology and Medical Biology of the University Medical Center Groningen (UMCG) for interpretation of immunohistochemistry images; Dr. Elisabeth Brouwer from the Department of Rheumatology and Clinical Immunology of the UMCG for sharing tissues for immunohistochemistry; Dr. Ana Jimenez Joaquina, Dr. André Nadler, Dr. Timothy Springer, Dr. Tobias Meyer, Dr. Michael Yaffe, Dr. Michael Davidson, and Dr. Aurélien Roux for plasmids.

GvdB has received funding from the European Research Council (ERC) under the European Union's Horizon 2020 research and innovation programme (grant agreement no. 862137). Open Access funding provided by the University of Groningen.

Author contributions: F.C. Stempels: conceptualization, methodology, investigation, writing of original draft, data curation; H.M. Warner: conceptualization, methodology, investigation; G. van den Bogaart: conceptualization, methodology, writing of original draft, supervision, project administration; M. Jiang, M. Moser, M.H. Janssens, S. Maassen, I.H. Nelen, R. de Boer, W.F. Jiemy, D. Knight, J. Selley, R. O'Cualain, M.V. Baranov, T.C.Q. Burgers, R. Sansevrino, D. Milovanovic, P. Heeringa, M.C. Jones, R. Vlijm, M. ter Beest: investigation. All authors participated in the writing, reviewing, and editing.

Disclosures: The authors declare no competing interests exist.

Submitted: 31 May 2022

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

# Supplemental material

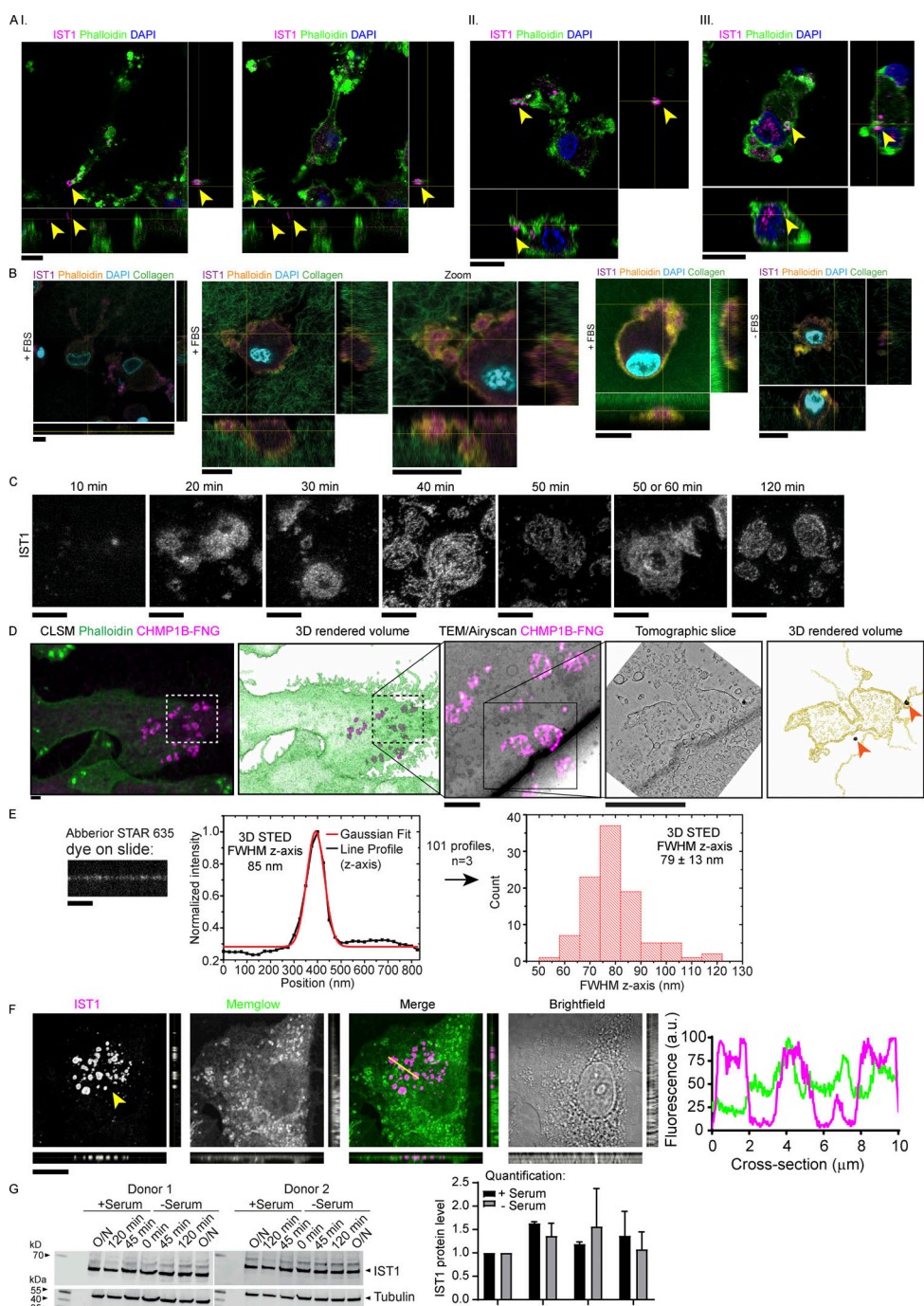

Figure S1. **IST1 structures are located at the plasma membrane of cells cultured on glass support and in collagen matrices. (A)** moDCs seeded on top of collagen matrix. Cells were allowed to migrate into the matrix for 5 h. Shown are confocal z-stacks. Cells are labeled for IST1 (magenta). Green: phalloidin. Blue: DAPI. The cells depicted in panels I-III are shown in Videos 2, 3, and 4, respectively. Scale bars: 10 μm. **(B)** Same as A, but now with FITC-labeled collagen (green in merge) and phalloidin in yellow. Scale bars, 10 μm. **(C)** STED imaging of moDCs cultured for the indicated times and immunostained for IST1. Scale bars: 1 μm. **(D)** Additional images of the cell shown in main Fig. 1 F. Top images: Confocal laser scanning microscopy (CLSM) images of a moDC cultured on glass support labeled with fluoronanogold (FNG) for CHMP1B (magenta). Green: phalloidin. Bottom images: CLEM using fluoronanogold labeling of CHMP1B (magenta hot) showing the area marked by the dotted lines in the top images. A tomography series of the 100-nm-thick bottom section (closest to the glass support) was generated. The bottom slice of this tomogram of the bottom section, and the 3D-rendered volume of all tomographic slices of this section are shown. Silver enhanced gold particles are indicated with an arrow. Scale bars: 2 μm. **(E)** Abberior STAR 635 dye was mounted on a coverslip and measured with identical 3D STED settings as used for measuring IST1 structures in main Fig. 1 G. A FWHM of 79 ± 13 nm (±1 STD; total of 40 measurements; n = 3) was obtained. The FWHM of the IST1 structure is thus ~24 nm thicker than just plain dye on a coverslip. Scale bar: 1 μm. **(F)** Confocal micrograph with orthogonal views of moDC immunolabeled for IST1 (magenta in merge) and the fluorescent lipophilic membrane marker Memglow (green). Line profiles of the cross-section indicated by the yellow line are shown. Scale bar: 10 μm. **(G)** Western blot showing IST1 expression in moDCS cultured on glass in the presence or absence of fetal bovine serum over time. The graph shows the mean ± SD of IST1 levels normalized to tubulin and to 0 min incubation. N = 2 donors. O/N: overnight. Source data are available for this figure: SourceData FS1.

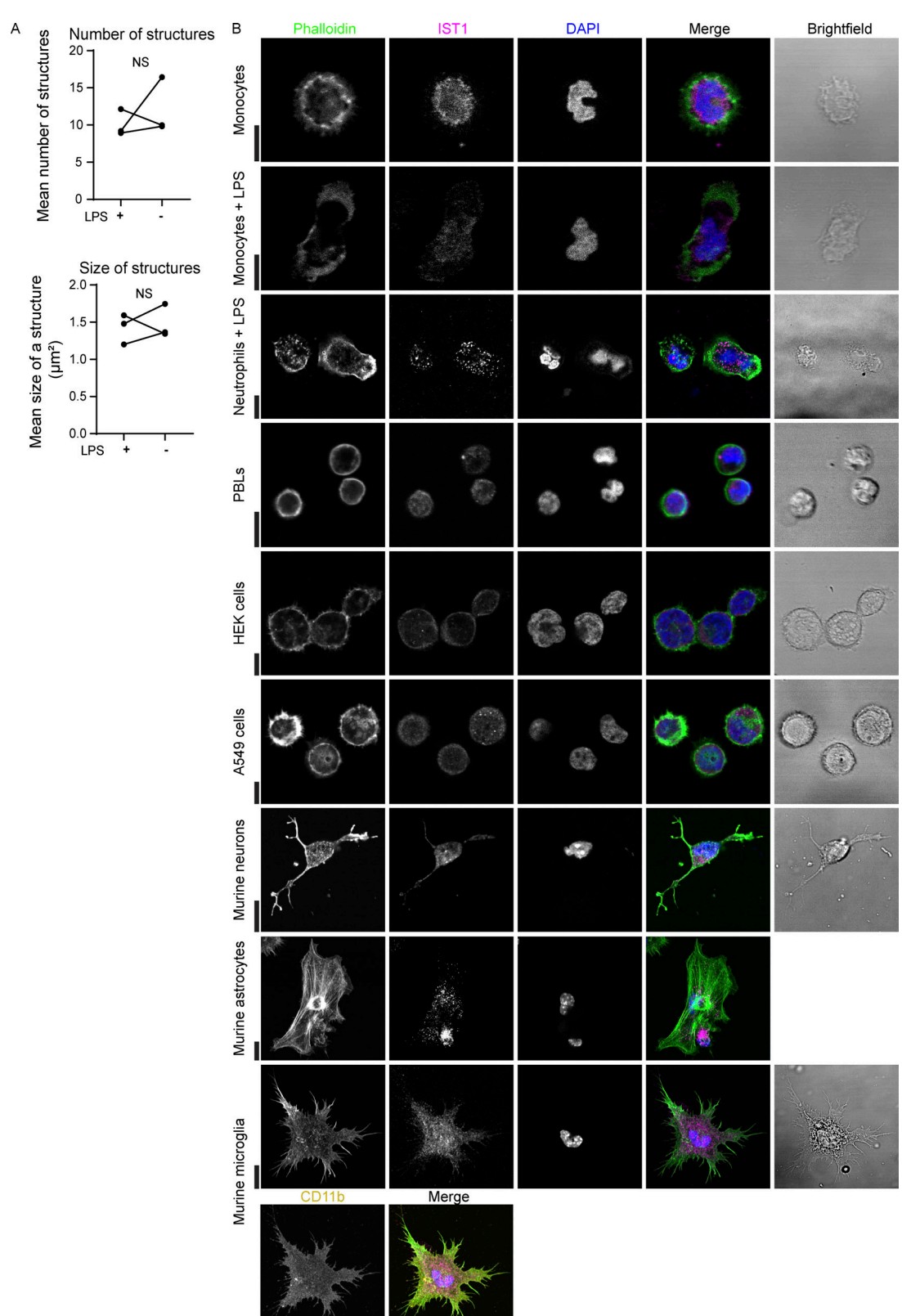

Figure S2. **LPS does not influence the IST1 structures and they are not present in non-migratory non-adhering cell types. (A)** moDCs were cultured in the presence of absence of 0.1 µg/ml LPS for 4 h and immunolabeled for IST1. Graphs show the average number of IST1-positive structures per cell and the average size of the structures. Each dot-line-dot pair represents a donor ($n$ = 3; paired two-sided $t$ tests were performed; NS = not significant; data distribution was assumed to be normal, but this was not formally tested). **(B)** Confocal micrographs of the indicated cell types immunolabeled for IST1 (magenta). A z-plane close to the coverslip is shown. Green: phalloidin. Blue: DAPI. Murine neurons, microglia and astrocytes were cultured for 6 h, whereas the other cells were cultured for 2 h. Murine microglia were also immunolabeled for the microglia marker CD11b (yellow). Scale bars: 10 µm.

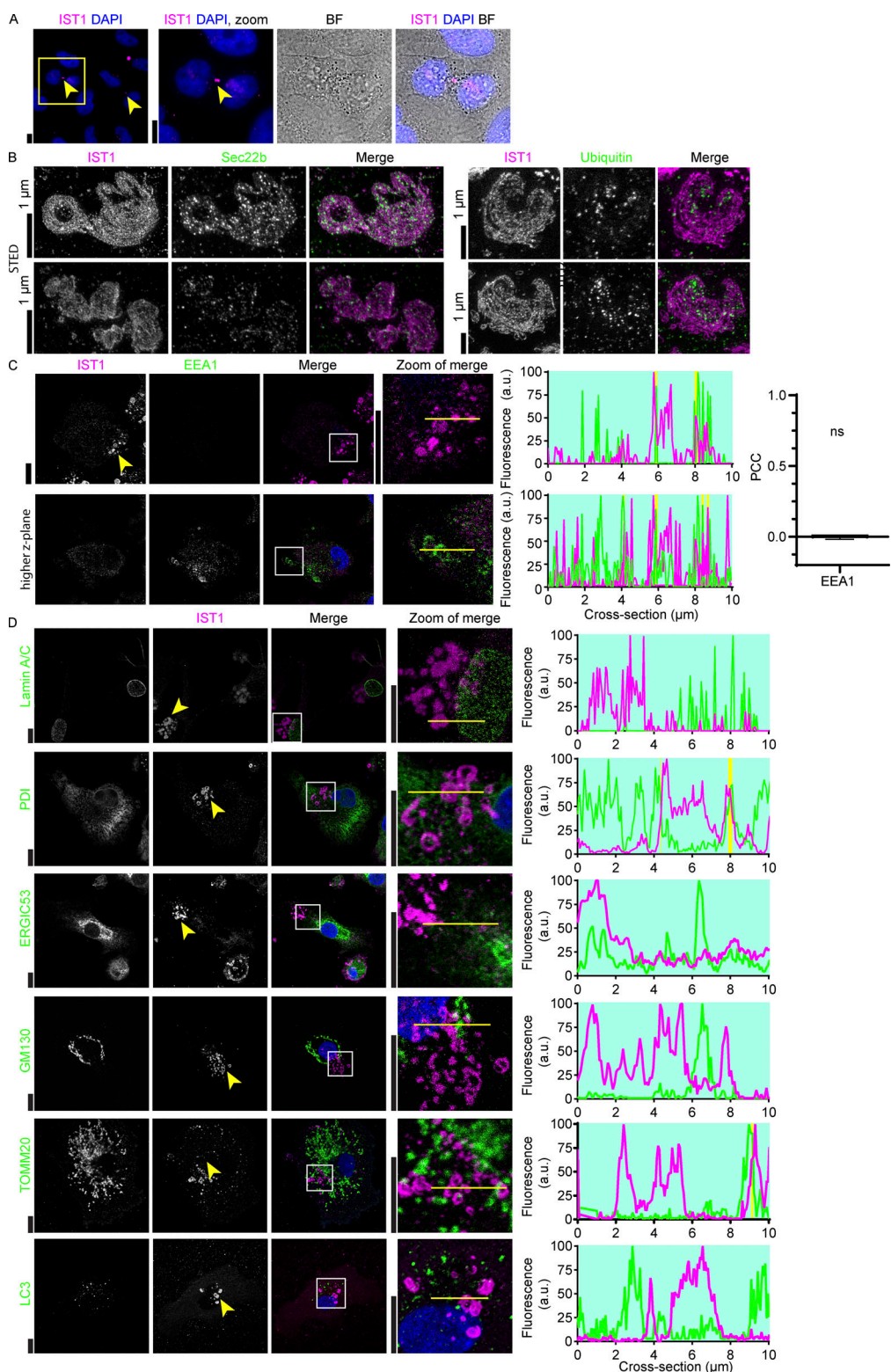

Figure S3. **Localization of IST1 structures with respect to organelle-markers. (A)** Dividing HeLa cells were labeled using rabbit anti-IST1 IgG (magenta) as a means to validate the antibody, because IST1 is known to be present in the midbody. IST1-positive midbodies are indicated with arrows. Blue: DAPI. Scale bars: 10 μm. **(B)** Stimulated emission depletion microscopy (STED) image of moDCs immunolabeled for IST1 (magenta in merge) and ubiquitin or Sec22b (green). Scale bars: 1 μm. **(C)** Confocal micrographs of moDCs immunolabeled for IST1 (magenta) and the endosomal marker EEA1 (green). Two different z-planes are shown, to show that EEA1 labeling was successful but EEA1 is not present in the IST1-positive structures. The bar graph shows the PCC ± SD (n = 3 donors). The PCC values were compared to 0 using a sample t test. Blue: DAPI. NS: not significant. For the statistical analysis, data distribution was assumed to be normal, but this was not formally tested. Scale bars: 10 μm. **(D)** Same as C, but now labeled for the nuclear envelope (Lamin A/C; green), endoplasmic reticulum (PDI), the ERGIC (ERGIC53), the Golgi complex (GM130), mitochondria (TOMM20), or autophagosomes (LC3). Scale bars: 10 μm.

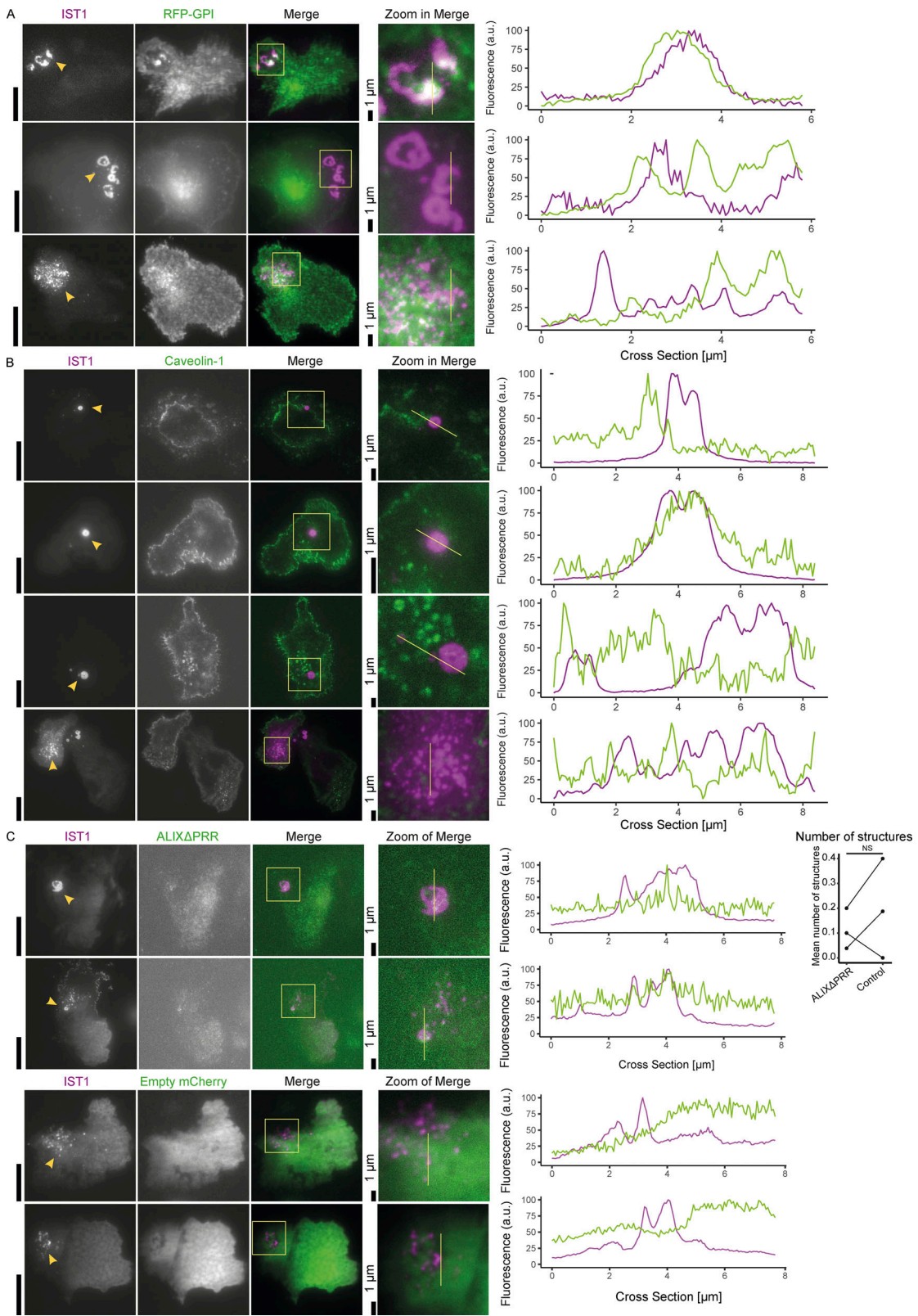

Figure S4. **IST1 structures do not colocalize with caveolin-1 and the ALIXΔPRR mutant. (A)** More examples for the GPI-anchored RFP shown in main Fig. 6 B. TIRF microscopy of moDCs co-expressing GFP-labeled IST (magenta in merge) with GPI-anchored RFP (green). **(B)** TIRF microscopy of moDCs co-expressing GFP-labeled IST (magenta in merge) with mCherry-tagged caveolin 1 (green). **(C)** TIRF microscopy of moDCs co-expressing GFP-labeled IST (magenta) with mCherry-tagged ALIXΔPRR mutant (green) or only mCherry (control). Graph shows the average number of IST1-positive structures for cells with detectable expression of both constructs for three different donors (paired two-sided *t* test). For the statistical analysis, data distribution was assumed to be normal, but this was not formally tested. Scale bars: 10 µm, unless indicated otherwise.

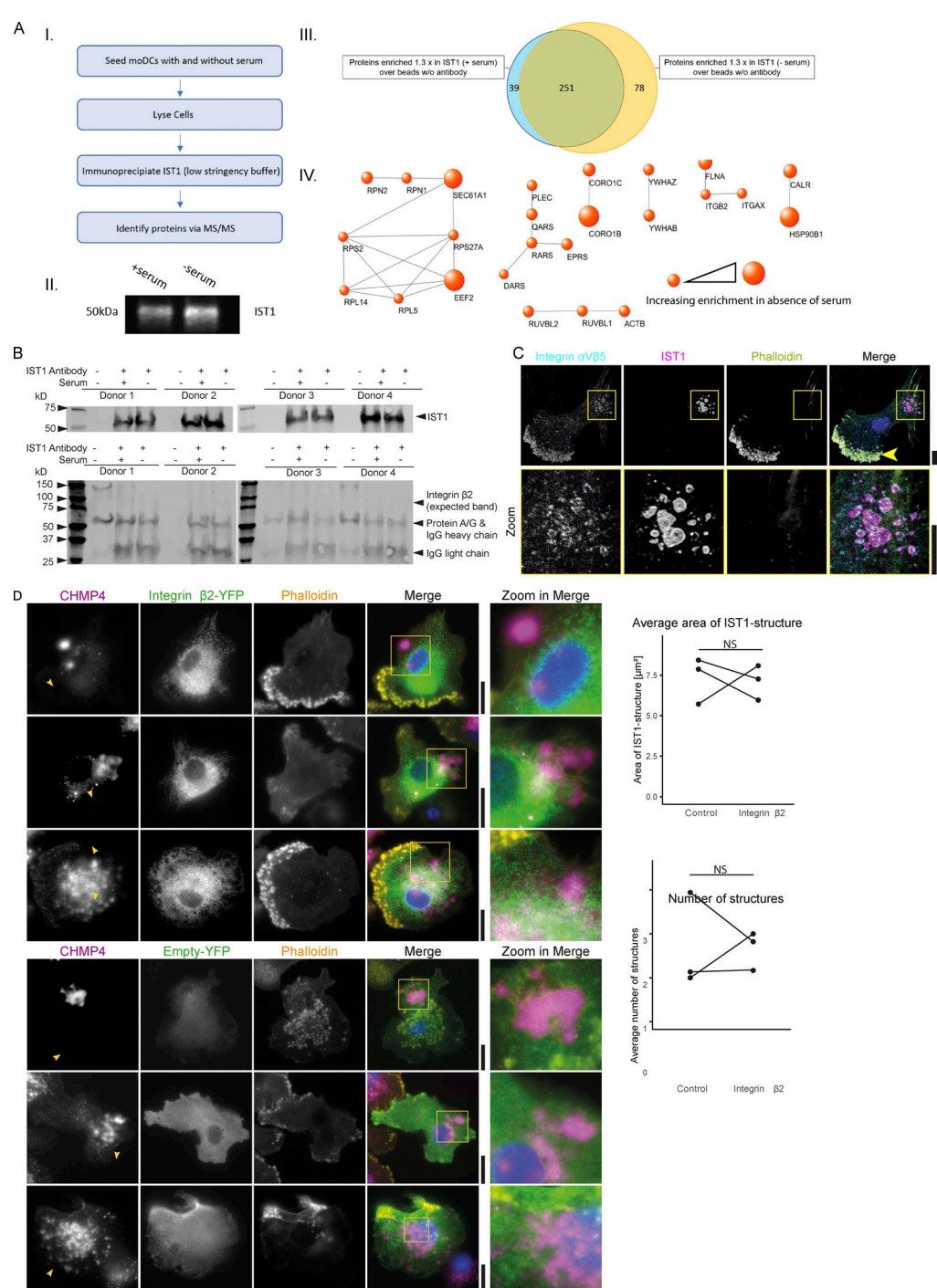

Figure S5.  **IST1-immunoprecipitation and proteomics. (A)** moDCs were cultured either in the presence or absence of serum followed by IST1 immuno-precipitation. Proteins were identified by mass spectrometry. Data is obtained using four donors. A.I: Flowchart of the IST1 IP. A.II: Western blot of eluted IP sample probed for IST1. A.III: Venn diagram showing proteins consistently enriched ≥1.3-fold in comparison with the sample with beads only (no antibody). The blue circle represents proteins upregulated in the samples cultured with serum and the yellow circle represents proteins upregulated in the samples cultured without serum. A.IV: Protein network showing proteins ≥1.3 times enriched in the serum-free condition over the serum-containing condition. Using the STRING tool, set at a confidence level of 0.9, proteins associated with cell adhesion were selected. Size of the dots corresponds with enrichment of the protein in the absence of serum. **(B)** Western blot of the pulldown confirming immunoprecipitation of the target IST1. Interaction with integrin β2 could not be detected by Western blot (expected band missing). We could not exclude that the IST1 antibody binds to IgG heavy chain, as IST1 runs at nearly the same height as the heavy chain antibody (and protein A/G). However, we confirmed the presence of IST1 with mass spec in A using the exact same procedures. **(C)** Confocal micrographs of moDCs immunolabeled for integrin αVβ5 (cyan in merger), IST1 (magenta) and phalloidin (yellow). For this integrin, we only observed co-localization with the ESCRT structures in one out of four tested donors. Note that integrin αVβ5 also locates to podosomes (arrow). **(D)** TIRF microscopy of moDCs co-expressing mCherry-labeled CHMP4 (magenta) with YFP-tagged integrin β2 (green) or only YFP (control). Graphs show the average size number of CHMP4-positive structures for cells with detectable expression of both constructs for three different donors. NS: not significant (paired two-sided *t* test; data distribution was assumed to be normal, but this was not formally tested). Scale bars: 10 μm. Source data are available for this figure: SourceData FS5.

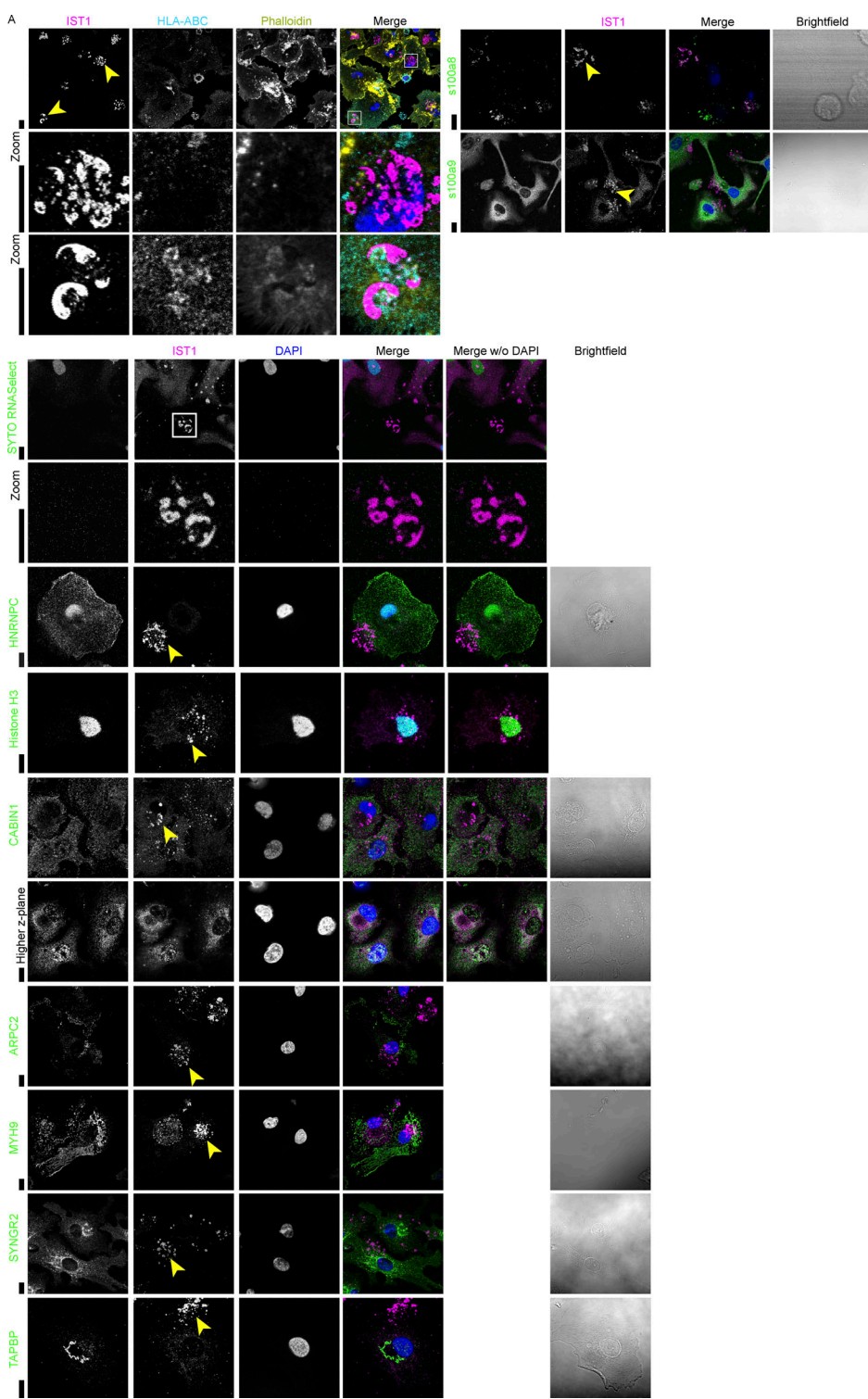

Figure S6.  **S100A8 locates at the IST1-structures. (A)** Confocal micrographs of moDCs immunolabeled for IST1 or CHMP1B (magenta) and a set of other proteins identified by mass spectrometry (green). The choice for which proteins were further investigated was based on the availability of antibodies and the function of the identified protein. Calprotectin subunit s100a8 localized in the IST1 structures, while the other subunit, s100a9, did not. The MHC class I form HLA-C did not co-localize with IST1 structures, except for one observation in one donor (bottom zoom), pointing at absent or very transient co-localization with IST1 structures. Neither the RNA stain SYTO RNASelect nor the DNA/RNA binding proteins Histone H3 and HNRNPC, nor the nuclear protein calcineurin-binding protein cabin-1 co-localized with IST1 structures. We also could not confirm the presence of cytoskeleton-involved proteins like actin related protein 2/3 complex subunit 2 (ARPC2), which is the actin-binding component of the Arp2/3 complex mediating actin polymerization, nor myosin-9, which plays a role in focal contact formation (Betapudi, 2010). Synaptogyrin-2, which may play a role in exocytosis in rat cells (Sugita et al., 1999), or tapasin, which is involved in the association of MHC class I with peptide and transporter associated with antigen processing (TAP), could also not be found in the structures. Proteins identified by the IST1-pulldown but not present in IST1-positive structures might bind IST1 in other subcellular locations. Scale bars: 10 µm.

**Figure S7. Latruculin B (LatB) control experiments and knockdowns of ESCRT proteins. (A)** moDC preincubated in the presence of 100 µM LatB and cultured for 1 h in the absence of LatB. The cell is immunolabeled for IST1 (magenta), phalloidin (green), and DAPI (blue). No F-actin staining was observed in the presence of LatB, and phalloidin showed weak nuclear staining. Scale bar: 10 µm. **(B)** Left: Scheme of workflow of adhesion isolation. Right: Volcano plot of mass spectrometry hits. Key consensus adhesion proteins (Horton et al., 2015) were depleted from adhesions in response to LatB treatment. **(C.I)** Western blot of moDCs treated with non-targeting (NT) siRNA, IST1 siRNA, or TSG101 siRNA. Total protein levels were measured using Ponceau S and the blot was probed for TSG101 and IST1. **(C.II)** Quantification of C.I. Protein levels were corrected for loading using Ponceau S intensities and subsequently normalized to the NT condition. Graphs show mean ± SD ($n ≥ 5$ donors). **(D)** Effect of IST1 knockdown and TSG101 knockdown on the average number of IST1-positive structures per cell, the average size of the structures and the average total area covered by the structures per cell. Effects are shown both in untreated cells and in LatB treated cells. Each dot-line-dot pair represents 1 donor. *: $P < 0.05$; NS: not significant (paired two-sided $t$ test). **(E)** Fold change in the number of adherent cells upon LatB treatment in moDCs with IST1 or TSG101 knockdown compared to cells treated with NT siRNA. (paired two-sided $t$ test). **(F)** Quantification of integrin β2 signal in the ESCRT structures, based on immunofluorescence labeling of moDCs for integrin β2 and IST1. Knockdown of IST1 and TSG101 significantly reduced the level of integrin β2 ($n = 3$ donors; paired two-sided $t$ test; ***: $P < 0.001$). **(G.I)** Western blot and quantification of knockdown of TSG101 and ALIX. Note unsuccessful knockdown of ALIX. **(G.II)** Flow cytometry of surface levels of integrin αM and integrin β2 in moDCs. TSG101 knockdown significantly lowered surface levels of integrin β2 ($n = 3$ donors; paired two-sided $t$ test). A representative histogram is shown. **(H)** DAPI influx in moDCs was not affected by knockdown of TSG101. Scale bars, 10 µm. For all statistical analysis, data distribution was assumed to be normal, but this was not formally tested. Source data are available for this figure: SourceData FS7.

Video 1.   **3D confocal microscopy reconstructions of moDC cultured in 3D collagen matrix for 5 h, showing IST1 structures at the plasma membrane.** IST1 is shown in magenta, phalloidin is shown in green, and DAPI in blue. Individual micrographs are shown in main Fig. 1 B.

Video 2.   **3D confocal microscopy reconstructions of moDC cultured in 3D collagen matrix for 5 h, showing IST1 structures at the plasma membrane.** IST1 is shown in magenta, phalloidin is shown in green, and DAPI in blue. Individual micrographs are shown in Fig. S1 A.

Video 3.   **3D confocal microscopy reconstructions of moDC cultured in 3D collagen matrix for 5 h, showing IST1 structures at the plasma membrane.** IST1 is shown in magenta, phalloidin is shown in green, and DAPI in blue. Individual micrographs are shown in Fig. S1 A.

Video 4.   **3D confocal microscopy reconstructions of moDC cultured in 3D collagen matrix for 5 h, showing IST1 structures at the plasma membrane.** IST1 is shown in magenta, phalloidin is shown in green, and DAPI in blue. Individual micrographs are shown in Fig. S1 A.

Video 5.   **Time-lapse TIRF microscopy.** Movie showing an moDC transfected with IST1-GFP (orange) with immobile IST1-positive structures over a time period of 27 min (incubated at RT). The movie was recorded immediately after seeding the cells at very low laser power (to limit phototoxicity). Frames were recorded with 3-s interval. Note the formation of the IST1 structures within ~5 min of seeding.

Video 6.   **Time-lapse TIRF microscopy.** Movie showing an mmoDC transfected with IST1-GFP (green) with immobile IST1-positive structures over a time period of 3 h and 10 min (incubated at RT). Frames were recorded with 19 min interval. Stills of this movie are shown in main Fig. 1 D.

Video 7.   **Time-lapse TIRF microscopy.** Movie showing an moDC transfected with IST1-GFP (orange) with immobile IST1-positive structures over a time period of 25 min, recorded with high laser power. Frames were recorded with 10 s interval. MoDCs are highly sensitive to phototoxicity, and the cell shows blebbing due to the disconnection of the plasma membrane from the cortical cytoskeleton. Note that although the IST1 signal diminishes (likely due to photobleaching), the structures persist upon cells death.

**Provided online are Data S1 and Table S1. Data S1 shows the sequence of the IST1-GFP construct and of the Fiji macro for the analysis of the number and size of the ESCRT structures. Table S1 shows IST1 pulldown mass spectrometry results.**

