## [Peer Review File · The Journal of Cell Biology]

Giant worm-shaped ESCRT-scaffolds surround actin-independent integrin clusters

Femmy Stempels, Muwei Jiang, Harry Warner, Magda-Lena Moser, Maaike Janssens, Sjors Maassen, Iris Nelen, Rinse de Boer, William Jiemy, David Knight, Julian Selley, Ronan O'Cualain, Maksim Baranov, Thomas Burgers, Roberto Sansevrino, Dragomir Milovanovic, Peter Heeringa, Matthew Jones, Rifka Vlijm, Martin ter Beest, and Geert van den Bogaart

Corresponding Author(s): Geert van den Bogaart, University of Groningen and Geert van den Bogaart, University of Groningen

Review Timeline:	Submission Date:	2022-05-31
	Editorial Decision:	2022-08-02
	Revision Received:	2023-01-25
	Editorial Decision:	2023-03-07
	Revision Received:	2023-03-23

Monitoring Editor: Johanna Ivaska

Scientific Editor: Dan Simon

Transaction Report:

DOI: <https://doi.org/10.1083/jcb.202205130>

August 2, 2022

Re: JCB manuscript #202205130

Prof. Geert van den Bogaart
University of Groningen
Molecular Immunology
Nijenborgh 9
9747AG
Netherlands

Dear Prof. van den Bogaart,

Thank you for submitting your manuscript entitled "Giant worm-shaped ESCRT-scaffolds support actin-independent integrin clusters." The manuscript was assessed by expert reviewers, whose comments are appended to this letter. In the reviews from three well-established experts in fields that overlapped the findings reported in this manuscript, there were some significant concerns that need to be addressed. However, our overall conclusion is that this study could be potentially appropriate for publication in JCB if appropriately revised. Consequently, we invite you to submit a revision if you can address the reviewers' concerns.

A particularly significant question is the contradiction between the proposed protective function and increased PM leakage in the presence of large ESCRT structures (Reviewers 2 & 3). In addition, all reviewers pointed out that the lack of functional data limits the impact of the study. From an editorial point of view, inclusion of additional functional experiments would make the paper stronger and would alleviate the concerns raised by the reviewers. Reviewer 1 finds the manuscript unclear and in need of significant clarification and re-writing of the text. Finally, all reviewers outline additional experimental concerns that seem more straightforward to address.

Please let us know if you are able to address the major issues outlined above and wish to submit a revised manuscript to JCB. Although a substantial amount of additional experimental data may be needed, we strongly encourage you to make every effort to resolve the concerns of these conscientious reviewers.

The typical timeframe for revisions is three to four months. While most universities and institutes have reopened labs and allowed researchers to begin working at nearly pre-pandemic levels, we at JCB realize that the lingering effects of the COVID-19 pandemic may still be impacting some aspects of your work, including the acquisition of equipment and reagents. Therefore, if you anticipate any difficulties in meeting this aforementioned revision time limit, please contact us and we can work with you to find an appropriate time frame for resubmission. Please note that papers are generally considered through only one revision cycle, so any revised manuscript will likely be either accepted or rejected.

GENERAL GUIDELINES:

Text limits: Character count for a Report is < 20,000; a full Research Article is < 40,000, not including spaces. Count includes title page, abstract, introduction, the joint Results & Discussion, and acknowledgments. Count does not include materials and methods, figure legends, references, tables, or supplemental legends.

Figures: A Report may include up to 5 main text figures; a full Research Article may have up to 10 main text figures. To avoid delays in production, figures must be prepared according to the policies outlined in our Instructions to Authors, under Data Presentation, <https://jcb.rupress.org/site/misc/ifora.xhtml>. All figures in accepted manuscripts will be screened prior to publication.

Supplemental information: There are strict limits on the allowable amount of supplemental data. Reports may have up to 3 supplemental figures; a full Research Article may have up to 5 supplemental figures. Up to 10 supplemental videos or flash animations are allowed. A summary of all supplemental material should appear at the end of the Materials and methods section.

Please note that JCB now requires authors to submit Source Data used to generate figures containing gels and Western blots with all revised manuscripts. This Source Data consists of fully uncropped and unprocessed images for each gel/blot displayed

in the main and supplemental figures. Since your paper includes cropped gel and/or blot images, please be sure to provide one Source Data file for each figure that contains gels and/or blots along with your revised manuscript files. File names for Source Data figures should be alphanumeric without any spaces or special characters (i.e., SourceDataF#, where F# refers to the associated main figure number or SourceDataFS# for those associated with Supplementary figures). The lanes of the gels/blots should be labeled as they are in the associated figure, the place where cropping was applied should be marked (with a box), and molecular weight/size standards should be labeled wherever possible. Source Data files will be made available to reviewers during evaluation of revised manuscripts and, if your paper is eventually published in JCB, the files will be directly linked to specific figures in the published article.

If you choose to resubmit, please include a cover letter addressing the reviewers' comments point by point. Please also highlight all changes in the text of the manuscript.

Regardless of how you choose to proceed, we hope that the comments below will prove constructive as your work progresses. We would be happy to discuss them further once you've had a chance to consider the points raised. You can contact the journal office with any questions, cellbio@rockefeller.edu or call (212) 327-8588.

Thank you for thinking of JCB as an appropriate place to publish your work.

Sincerely,

Johanna Ivaska, PhD
Monitoring Editor
Journal of Cell Biology

Dan Simon, PhD
Scientific Editor
Journal of Cell Biology

Reviewer #1 (Comments to the Authors (Required)):

This manuscript reports the existence of previously undescribed giant assemblies of ESCRT components at the plasma membrane. The proposed role of these structures is to protect a challenged plasma membrane from leakiness. The authors conducted an impressive breath of experiments, using state-of-the-art technologies in order to characterize these structures. The potential functions of such structures in cell adhesion and/or membrane protection are extremely exciting but the authors did not test directly their own hypotheses.

As a general comment, the paper is extremely difficult to read, mostly because the description of the results is very poor and it is often not clear what the authors measured exactly. For example, I do not understand how the authors come to the conclusion that IST1 structures are found at membrane "deformations". The reader is referred to figures describing the thickness of IST1 labelling versus a glass labeling. There is no explanation as to what this thickness means, and I am not sure at all that it means anything anyway. This level of description does not allow to determine whether the conclusions are indeed supported by the data, because it is difficult to understand what the data means.

Specific points:

- Description of the structures. It is stunning that, regarding the very strong enrichment in ESCRT components, these structures were never reported before. This may be due to their transient nature upon cell adhesion. But authors need to comment on that. Also, the shape and distribution of these structures is highly variable, sometimes frankly looking like round-shaped late endosomes and sometimes looking like clathrin-coated plaques/reticular adhesions. Along this line, although the authors checked many different organelle markers, they should really stain for β 5-integrin, in cell types expressing it, because this is the best marker of these types of adhesions. Also, the interactions of ESCRT structures with the substrate is not properly documented. First, in 3D setups, it is not clear at all that the IST1 accumulations correspond to what is seen in 2D. The authors need to check whether ESCRT structures are also in contact with the substrate in 3D and thus, they need to visualize collagen fibers. Second, even in the 2D conditions, interaction with the substrate could be more directly documented by performing internal reflection microscopy for example, and/or by showing a video in which IST1 structures are left behind a migrating cell. The extreme stability of these structures relative to the substrate is probably the best evidence of a connection with the substrate but the IST1-GFP video (video 5) shows a variable pattern in different cells, and cells are dying/not moving.

- Functions. The authors propose two functions for these structures? The first one is cell adhesion. The authors dismissed this possibility based on the fact that ESCRT components-depletion does not impair the increased adhesion potential observed in latrunculin treated cells. The authors should perform proper cell adhesion assays to determine adhesion kinetics in WT versus

ESCRT-depleted cells in normal conditions. It seems that ESCRT structures are dispensable to organize integrin clusters, although these clusters are often found at the center of an ESCRT ring. Is it possible that integrin clustering is transient and that it may escape scrutiny in fixed samples? Also, are integrins required for the formation of ESCRT assemblies?

The second hypothesis is that ESCRT structures serve as a protective layer for the plasma membrane. The only experiment proposed to test this hypothesis is the DAPI leakiness assay. First, I am not sure how standard is this assay and there are other methods that should complement this one. Second, the authors only compare WT cells in starved versus serum conditions. This is merely a correlation with the starvation-dependent formation of ESCRT structures. The authors need to test directly the role of ESCRT components in their own hypothesis.

Minor points:

- In Supp. Fig5A, phalloidine staining seems to accumulate in the center of ESCRT structures. There is no mention of this in the text.
- A scheme would help to understand how the authors put together all their data in a coherent working model

Reviewer #2 (Comments to the Authors (Required)):

The article by Stempels and colleagues describes a new structure formed most probably at the plasma membrane by ESCRT-III polymers in immunological cells that diffuse in tissues. This structure is anti-correlated with actin-depend adhesion domains. The authors clearly show that:

- the large structures (several microns) contains several ESCRT-III subunits
- they are associated with specific integrins involved in actin-independent adhesion sites
- They are found essentially in migrating cells and tissue penetrating cells.

The article is very well conducted, using the many tools of cell biology to characterize the structures in details. The characterization is quite extensive and convincing, making the article a clear candidate for publication in Journal of Cell Biology.

I only have a few comments that could be addressed in a revised manuscript.

1-I think the membrane association of the ESCRT-III structure is not fully characterized. It would be helpful to see if the presence of these large ESCRT-III structures induces any specific domain within the membrane, or curvature. Using a membrane marker such as cellmask, or GFP-CAAX could be helpful. I think the structure may be flat, but vesicular budding or curvature may be observed in the center or at the rim of the ESCRT-III structures.

2-Use of protein domains binding to specific lipids could be helpful to know if the structures induce a lipid segregation. PIP-binding domains such as the FYVE or the PH domain would be helpful. Also, the GFP-GPI could be a good idea to see if the liquid ordered phase is either excluded or present under the ESCRT-III structure.

3-While the hypothesis of ESCRT-III coats around actin-independent adhesion sites would prevent membrane tearing, I am a bit confused by the experiment presented in lines 11-15 page 6. The authors found more leakage when ESCRT-III structures are present (serum-free), while ESCRT-III presence or absence does not affect actin-independent adhesion. Isn't this result contradictory with the author's hypothesis? I would have expected that more leakage is seen when ESCRT-III structures are absent.

4-I find very interesting that several Exosome markers are found in the center of the ring-like ESCRT-III structures, in particular CD63, and ubiquitin. I am thus wondering if these structures could be focalizing platforms for exosome budding, which have been shown to have essential signaling roles in immune response. This is also supported by the fact that these structures are sometimes observed outside cells (striking in fig 1A).I think the authors should look at other classical markers such as Alix, and tetraspanins. I would recommend to overexpress Alix-DeltaPRD, which stimulates one exosome pathway (Larios et al. JCB 2020) and leads to enrichment of tetraspanins in the exosomal fraction, to see if it regulates the size and the number of ESCRT-III structures.

5-I was wondering if overexpression of actin-independent integrins could increase the number/size of ESCRT-III structures. As a control, overexpression of integrins that are linked to actin should be tested.

Reviewer #3 (Comments to the Authors (Required)):

In this manuscript, Stempels et al. have shown that ring-shaped ESCRT structures consisting of IST1 and other ESCRT-III proteins are formed around actin-independent integrin clusters at the plasma membrane. The authors claim that these ESCRT structures are required to protect the plasma membrane from damage/rupture at the sites of cell adhesion via the integrin

clusters. However, some aspects are still less convincing and requires more experiments to be confirmed.

Minor revisions:

In page 4 line 10: I think it should be supplementary figure 3B.

Major revisions:

In Fig 2C: It is necessary to stain for integrins in the temporal artery biopsy of GCA. This would further correlate the migratory phenotype of CD68+ and vimentin+ fibroblasts to the presence of integrin clusters at the plasma membrane. In addition, it adds to the manuscript an observation related to a pathological state (e.g., inflammation, cancer, etc..).

In Fig 3: A co-immunoprecipitation experiment should also be performed to show: (i) the potential interaction between IST1 and integrins, (ii) The absence of actin in these integrin clusters surrounded by IST1. This could be obtained by immunoprecipitating IST1 and blotting for integrins and F-actin.

The authors have downregulated IST1 and Tsg101 expression by siRNA. It is well known that the depletion of Tsg101 disrupts receptor trafficking and increases receptor density at the plasma membrane. It would be also important to measure the total integrin receptor levels by immuno-blotting and the migration rate of adherent cells (e.g., HEK 293 cells). In addition, it is necessary to find out how the integrin clusters are affected by IST1 depletion.

In Fig 4C: The experimental interpretation is unclear. It is said that in the absence of serum the ESCRT structures are formed. Thus, how it is interpreted that the leakage of membrane impermeable DAPI into the cells increased upon absence of serum. If these ESCRT structures are protecting or aiding in membrane repair, it shouldn't decrease the DAPI leakage?

It would also be more convincing to treat cells with certain chemotherapeutic drugs or any other chemical compounds known to induce mild ruptures in the plasma membrane and to show whether the recruitment of these ESCRT structures to the plasma membrane is being altered.

Reviewer #1:

This manuscript reports the existence of previously undescribed giant assemblies of ESCRT components at the plasma membrane. The proposed role of these structures is to protect a challenged plasma membrane from leakiness. The authors conducted an impressive breath of experiments, using state-of-the-art technologies in order to characterize these structures. The potential functions of such structures in cell adhesion and/or membrane protection are extremely exciting but the authors did not test directly their own hypotheses.

1.1 *As a general comment, the paper is extremely difficult to read, mostly because the description of the results is very poor and it is often not clear what the authors measured exactly. For example, I do not understand how the authors come to the conclusion that IST1 structures are found at membrane "deformations". The reader is referred to figures describing the thickness of IST1 labelling versus a glass labeling. There is no explanation as to what this thickness means, and I am not sure at all that it means anything anyway. This level of description does not allow to determine whether the conclusions are indeed supported by the data, because it is difficult to understand what the data means.*

- We apologize for the confusion and have extensively revised the text throughout the manuscript to make it clearer. Specifically, the conclusion that the structures locate to membrane deformations is based on correlative light-electron microscopy (CLEM), shown in main Figure 2C and Supplementary Figure 1F (formerly Supplementary Figure 1C, which the reviewer may have confused for former Supplementary Figure 1BII on the thickness of the structures). The figures describing the thickness of the IST1 labeling are main Figure 2B and Supplementary Figure 1E. These figures show that the structures are ~24 thick, as the axial full-width at half-maximum (FWHM) intensity of the STED microscopy of the IST1 structures (main Figure 2B) was 24 nm larger compared to just plain dye on a coverslip (Supplementary Figure 1E). We have revised the text to better explain this (*Results* section, page 3).

1.2 *Description of the structures. It is stunning that, regarding the very strong enrichment in ESCRT components, these structures were never reported before. This may be due to their transient nature upon cell adhesion. But authors need to comment on that. Also, the shape and distribution of these structures is highly variable, sometimes frankly looking like round-shaped late endosomes and sometimes looking like clathrin-coated plaques/reticular adhesions. Along this line, although the authors checked many different organelle markers, they should really stain for $\beta 5$ -integrin, in cell types expressing it, because this is the best marker of these types of adhesions. Also, the interactions of ESCRT structures with the substrate is not properly documented. First, in 3D setups, it is not clear at all that the IST1 accumulations correspond to what is seen in 2D. The authors need to check whether ESCRT structures are also in contact with the substrate in 3D and thus, they need to visualize collagen fibers. Second, even in the 2D conditions, interaction with the substrate could be more directly documented by performing internal reflection microscopy for example, and/or by showing a video in which IST1 structures are left behind a migrating cell. The extreme stability of these structures relative to the substrate is probably the best evidence of a connection with the substrate but the IST1-GFP video (video 5) shows a variable pattern in different cells, and cells are dying/not moving.*

- As requested, we now remark in the *Discussion* section on page 8 of the revised manuscript on possible reasons why the ESCRT structures have not been observed previously. We do not think that this is due to their transient nature, as the structures are highly stable (*i.e.*, present for hours after formation, see for example main Figure 5C and Movie 4). Instead, we believe that the function of ESCRT proteins has mainly been studied in cell lines that do not form these structures. As described in the manuscript, we only observed them in migratory tissue-infiltrating cell types (*i.e.*, macrophages, DCs, fibroblasts; main Figure 4A and Supplementary Figure 2C).
- We also comment in the *Results* section on page 3 on the heterogeneity of the shapes. Mostly, we observed clusters of structures with irregular ring or worm-shaped structures.
- As requested, we performed immunofluorescence labelling for integrin $\beta 5$. However, we only observed co-localization for one out of four donors (see new Supplementary Figure 4C). As we do not observe NUMB, DAB2 and clathrin light chain (main Figure 7F), we do not think that we can conclude that the

ESCRT structures are reticular adhesions. Please also note that we observed the ESCRT structures in terminally differentiated (*i.e.*, non-dividing) cell types (*e.g.*, monocyte-derived dendritic cells and macrophages) whereas the role of reticular adhesions is most clear in cell division. We now discuss this in the *Results* section at page 6 of our revised manuscript.

- As requested, we have performed experiments with fluorescently-labelled collagen to show that the cells are surrounded by the collagen. These new data are presented in Supplementary Figure 1B-C.
- As requested, we have included new TIRF microscopy experiments showing that the IST1 structures are directly present at the glass support in 2D culture: In new main Figure 1D we show TIRF and epi-fluorescence side-by-side of IST-GFP together with Lysotracker. Note the increased signal-to-background of the structures confirming proper TIRF. In new Supplementary Movie 5, we show the formation of the IST1 structures using live-cell time-lapse TIRF microscopy. Please note that this was highly challenging, as monocyte-derived DCs are highly sensitive to phototoxicity and we had to use very low light exposure. In new Supplementary Movie 7, we show using TIRF microscopy that the IST1 structures persist upon cell death caused by this phototoxicity (blebbing: the disconnection of the F-actin cytoskeleton from the membrane). Further supporting our conclusion that the ESCRT structures interact with the extracellular substrate is the finding that the IST1 structures persist upon removal of the cells by cold shock (a common procedure for harvesting monocyte-derived DCs; see main Figure 6C). We now discuss this in the *Results* section at page 3 of our revised manuscript.

1.3 *Functions. The authors propose two functions for these structures? The first one is cell adhesion. The authors dismissed this possibility based on the fact that ESCRT components-depletion does not impair the increased adhesion potential observed in latrunculin treated cells. The authors should perform proper cell adhesion assays to determine adhesion kinetics in WT versus ESCRT-depleted cells in normal conditions. It seems that ESCRT structures are dispensable to organize integrin clusters, although these clusters are often found at the center of an ESCRT ring. Is it possible that integrin clustering is transient and that it may escape scrutiny in fixed samples? Also, are integrins required for the formation of ESCRT assemblies?*

We apologize if this was unclear, but we do not think that the ESCRT structures are involved in the cell adhesion and/or the formation of the integrin clusters and we also do not think that the integrins are directly involved in the formation of the ESCRT structures. Instead, we think that the ESCRT structures repair or prevent membrane damage at actin-independent adhesion sites. This is based on the following evidence:

- Please note that not all integrin clusters were surrounded by ESCRT structures and the structures did not precisely locate at the sites of the integrin clusters (main Figure 7A). The large distance ($> \mu\text{m}$) between the ESCRT and integrins argues against a direct interaction between the ESCRT proteins and the integrins (see also Reviewer #3, point #3.3). Moreover, the ESCRT structures were visible almost immediately after seeding (see point #1.2; main Figure 5C and Movie 4), whereas their wrapping of the integrin clusters peaked much later (40 min, main Figure 7A). As we now explain at page 7 of our revised manuscript, these data are not in line with a role of the ESCRT proteins in the formation of the integrin clusters.
- We quantified the clusters of integrin $\beta 2$ in the IST1 and TSG101 knockdown cells (new Supplementary Figure 5F). We found that siRNA-knockdown of either of these proteins resulted in a small but significant reduction of the intensity of integrin $\beta 2$ clusters. Although this might point to a role of the ESCRT structures in the formation and/or recycling of integrin clusters, these data are not conclusive as ESCRT also plays roles in endosomal trafficking and this might very well relate to these or other roles of the ESCRT proteins. Indeed, we show by flow cytometry that knockdown of ESCRT proteins also lowers total surface levels of integrin $\beta 2$ in cells cultured in suspension (new Supplementary Figure 5G; see also reviewer #3, point #3.4 below). As we discuss in our revised manuscript (page 8), the multiple cellular functions of ESCRT make it extremely challenging to determine specific functional role of the ESCRT structures. These data are discussed in the *Results* section at page 8 and *Discussion* section at pages 8–9 of our revised manuscript.

- We also tested the effects of knockdown of ESCRT proteins on membrane leakage. Knockdown of TSG101 only increased DAPI influx for two out of three donors (new Supplementary Figure 5H), possibly due to the fact that a low level of TSG101 was still present and this level may suffice for recruitment of downstream ESCRT proteins. To overcome this challenge and directly determine whether the ESCRT structures formed at sites of membrane damage, we exposed the cells to silica crystals that are known to induce membrane damage [Beckwith, et al. 2020 *Nature Comm.* 11:2270]. We observed large ESCRT structures containing IST1, CHMP4A, CHMP4B and ALIX at the plasma membrane contact sites with silica crystals (new main Figure 8D and below). Live cell imaging showed that IST1-GFP was also recruited to these contact sites (new main Figure 8D). Please note that a role of ESCRT structures in membrane repair is in line with known roles of ESCRT proteins in repair of membrane damage inflicted by a laser or pore-forming agent [Jimenez, et al. 2014 *Science* 343:6174; Scheffer, et al. 2014 *Nature Comm.* 5:1]. This is discussed in the *Results* section at page 8 of our revised manuscript.
- In a fraction (<5%) of dendritic cells cultured in 3D collagen, we observed large ESCRT structures on the nuclear membrane (new Supplementary Figure 1C). Whereas we do not know whether this is related to damage of the nuclear membrane, this seems likely as the nuclear deformation caused by tumor cell migration in confining microenvironments, such as dense collagen matrices, is known to damage the nuclear membrane, and ESCRT proteins play a role in repair of the nuclear membrane [Denais, et al. 2016 *Science.* 352:353; Raab, et al. 2016 *Science.* 352:359]. This is discussed in the *Discussion* section at page 9 of our revised manuscript.
- To determine whether integrins are required for the formation of the ESCRT structures, we adhered monocyte-derived dendritic cells on poly-L-lysine coated cover glasses in the presence of an excess (5 mM) of the calcium chelator EDTA. Although the cells were rounded and less stretched, the ESCRT structures were still formed in the presence of EDTA and we observed no consistent changes in their size and number (new main Figure 7C). This is in line with our new data with overexpression of integrin $\beta 2$, also showing no apparent effects on the size and number of the ESCRT structures (new Supplementary Figure 4D; see also Reviewer #2, point #2.5 below). These new data are discussed in the *Results* section on page 6 of our revised manuscript.

1.4 *The second hypothesis is that ESCRT structures serve as a protective layer for the plasma membrane. The only experiment proposed to test this hypothesis is the DAPI leakiness assay. First, I am not sure how standard is this assay and there are other methods that should complement this one. Second, the authors only compare WT cells in starved versus serum conditions. This is merely a correlation with the starvation-dependent formation of ESCRT structures. The authors need to test directly the role of ESCRT components in their own hypothesis.*

- Concerning the DAPI leakage assay: Assays with DNA intercalating/staining dyes as membrane leakage markers have been used previously, for example in [Jimenez, et al. 2014 *Science* 343:6174; Corrotte, et al. 2016 *Methods Cell Biol.* 126:139; Ritter, et al. 2022 *Science* 376:6591]. We now mention this in the *Results* section on page 7 of our revised manuscript. Please see above (at point #1.3) regarding the involvement of the ESCRT structures in membrane repair.
- We did not find direct evidence that the ESCRT structures also protect the plasma membrane, and have weakened this claim throughout our manuscript. We now only speculate on this role in the last paragraph of the *Discussion* section (page 9). Moreover, we changed the title to “Giant worm-shaped ESCRT-scaffolds surround actin-independent integrin clusters”.

1.5 *In Supp. Fig5A, phalloidine staining seems to accumulate in the center of ESCRT structures. There is no mention of this in the text.*

- This comment is unclear. Supplementary Figure 5A shows no accumulation of phalloidin in the center of the ESCRT structures, but (weak) phalloidin staining in the nucleus in latrunculin-treated cells, likely due to non-specific binding of the phalloidin in absence of F-actin. We now comment on this on page 6 of our revised manuscript.

1.6 *A scheme would help to understand how the authors put together all their data in a coherent working model*

- As requested, we now provide a scheme with the proposed model (new main Figure 9, discussed in the Discussion section at page 9 of the revised manuscript).

Reviewer #2:

The article by Stempels and colleagues describes a new structure formed most probably at the plasma membrane by ESCRT-III polymers in immunological cells that diffuse in tissues. This structure is anti-correlated with actin-depend adhesion domains. The authors clearly show that:

- The large structures (several microns) contains several ESCRT-III subunits*
- They are associated with specific integrins involved in actin-independent adhesion sites*
- They are found essentially in migrating cells and tissue penetrating cells.*

The article is very well conducted, using the many tools of cell biology to characterize the structures in details. The characterization is quite extensive and convincing, making the article a clear candidate for publication in Journal of Cell Biology. I only have a few comments that could be addressed in a revised manuscript.

2.1 I think the membrane association of the ESCRT-III structure is not fully characterized. It would be helpful to see if the presence of these large ESCRT-III structures induces any specific domain within the membrane, or curvature. Using a membrane marker such as cellmask, or GFP-CAAX could be helpful. I think the structure may be flat, but vesicular budding or curvature may be observed in the center or at the rim of the ESCRT-III structures.

- Please see above at reviewer comment #1.2 regarding the evidence of the membrane association of the ESCRT structures. In addition, we have performed experiments with the membrane marker MemGlow confirming that the ESCRT structures are located at the plasma membrane (Supplementary Figure 1G). These experiments, and the 3D-STED experiments, showed that the structures are relatively flat. However, and as discussed above at reviewer comment #1.1, the CLEM data revealed membrane deformations at the ESCRT structures (main Figure 2C and Supplementary Figure 1F). Moreover, the absence of curvature in the MemGlow and STED experiments might be an artifact of the rigid glass support. These data are discussed at page 3 of our revised manuscript.

2.2 Use of protein domains binding to specific lipids could be helpful to know if the structures induce a lipid segregation. PIP-binding domains such as the FYVE or the PH domain would be helpful. Also, the GFP-GPI could be a good idea to see if the liquid ordered phase is either excluded or present under the ESCRT-III structure.

We have performed the requested experiments:

- Two-color TIRF microscopy with monocyte-derived dendritic cells co-expressing CHMP4B-mCherry with GFP-tagged PH-domain of PLC δ showed that the ESCRT structures are completely devoid of PI(4,5)P₂.
- Two-color TIRF microscopy with monocyte-derived dendritic cells co-expressing CHMP4B-mCherry with GFP-tagged PH domain of AKT showed that the ESCRT structures are also completely devoid of PI(3,4,5)P₃.
- For unclear reasons, monocyte-derived dendritic cells don not seem to tolerate overexpression of the FYVE domain of EAA1. To overcome this, we used the the PX-domain of p40^{phox} (NCF4), which is also specific for PI(3)P [Kanai et al., (2001) *Nat. Cell Biol.* 3:675]. Two-color TIRF microscopy showed that the ESCRT structures are also completely devoid of PI(3)P.
- Because the ESCRT protein TSG101 binds to PI(3,5)P₂ [Whitley et al., (2003) *JBC* 278:38786]. We also expressed the N-terminal sequence of MCOLN1, which is specific for PI(3,5)P₂ [Li et al., (2013) *PNAS* 110:21165]. Two-color TIRF microscopy showed that the ESCRT structures are also completely devoid of PI(3,5)P₂.

These data are shown in new main Figure 3B and new Supplementary Figure 2A and discussed at page 3–4 of our revised manuscript.

- We also performed two-color TIRF microscopy with monocyte-derived dendritic cells co-expressing GPI-anchored RFP with IST1-GFP. These experiments showed that clusters of GPI-anchored RFP were surrounded by the ESCRT structures (*i.e.*, not overlapping, but in the center of the rings; new main Figure 5B and new Supplementary Figure 3E). As GPI-anchored proteins are also cargoes of extracellular vesicles [Vidal. 2020 *Adv. Drug Deliv. Rev.*161:110], this might relate to the potential role of the ESCRT structures in the production of extracellular vesicles (see point #2.4 below). We discuss these new data at page 4 of our revised manuscript.

2.3 *While the hypothesis of ESCRT-III coats around actin-independent adhesion sites would prevent membrane tearing, I am a bit confused by the experiment presented in lines 11-15 page 6. The authors found more leakage when ESCRT-III structures are present (serum-free), while ESCRT-III presence or absence does not affect actin-independent adhesion. Isn't this result contradictory with the author's hypothesis? I would have expected that more leakage is seen when ESCRT-III structures are absent.*

- We apologize if this was unclear. We believe that the increased abundance of the ESCRT structures that we observed in the serum-free condition is caused by the increased membrane damage. We think that the ESCRT structures are formed to cope with this challenging environment, and that without these ESCRT structures, there would be even more leakage in the serum-free condition. However, knockdown of TSG101 only increased DAPI influx for two out of three donors, and had the opposite effect for the third donor (new Supplementary Figure 5H). The alternative (opposite) conclusion would be that the structures cause the leakage in the serum-free condition, but this is not in line with the functions of ESCRT proteins as described in literature. Please see also at reviewer #1 point #1.3 for further explanation of our new data supporting a role of the ESCRT structures in membrane repair.
- We have no direct evidence that the ESCRT structures also prevent membrane tearing. However, as the ESCRT structures are extremely stable and persist after cell death (see reviewer #1 comment #1.2), they might potentially also do this. We have revised the text throughout the manuscript to weaken this conclusion, including in the *Abstract*, *Introduction*, and *Results* sections, and now clearly state that this is an untested hypothesis in the *Discussion* section (page 9). We also changed the title to “Giant worm-shaped ESCRT-scaffolds surround actin-independent integrin clusters”.

2.4 *I find very interesting that several Exosome markers are found in the center of the ring-like ESCRT-III structures, in particular CD63, and ubiquitin. I am thus wondering if these structures could be focalizing platforms for exosome budding, which have been shown to have essential signaling roles in immune response. This is also supported by the fact that these structures are sometimes observed outside cells (striking in fig 1A). I think the authors should look at other classical markers such as Alix, and tetraspanins. I would recommend to overexpress Alix-DeltaPRD, which stimulates one exosome pathway (Larios et al. JCB 2020) and leads to enrichment of tetraspanins in the exosomal fraction, to see if it regulates the size and the number of ESCRT-III structures.*

- As requested, we stained for tetraspanins CD9 (new main Figure 5A), CD63 (main Figure 5A) and ALIX (main Figure 3A). We never observed CD9 with the ESCRT structures, but clusters of CD63 were present in the centers of ring-shaped ESCRT structures and ALIX co-localized with the ESCRT structures. In addition, we also observed clusters of GPI-anchored RFP in the centers of the ESCRT rings and GPI-anchored proteins are also found in extracellular vesicles (new main Figure 5B and new Supplementary Figure 3E; see point #2.2). We discuss these data in the Results section at page 4 of our revised manuscript.
- We agree with the reviewer that the ESCRT structures might be responsible for the formation of extracellular vesicles. However, we think that the formation of ectosomes (*i.e.*, shedding of the plasma membrane) is more likely than the formation of exosomes (secretion of multilamellar bodies), as the ESCRT structures are located at the plasma membrane and our data show that the ESCRT-structures dissociate from the cells as a whole. Starting from 1 hr after seeding, we observed patches of clustered structures that were increasingly no longer associated with the cells (main Figure 5C). These extracellular clusters were still adhered to the glass support and were surrounded by plasma membrane, as shown by HLA-DR positive areas surrounding the clusters (main Figure 5D). We now better

highlight the similarities between our ESCRT structures and characteristics of ectosome formation (*e.g.*, local degradation of the cytoskeleton, changes in lipid composition), and cite papers reporting ESCRT-mediated extracellular vesicle formation [Colombo, et al. 2013 *J. Cell Sci.* 126:5553; Larios, et al. 2020 *J. Cell Biol.* 219: e201904113; Nabhan, et al. 2012 *PNAS* 109: 4146; Wehman, et al. (2011) *Curr. Biol.* 21: 1951]. Thus, the ESCRT structures might mediate the formation of extracellular vesicles and/or fragments of plasma membrane, which could also have signaling functions (although this is an untested hypothesis). We now discuss this in the *Results* section at page 4–5 and the *Discussion* section at page 9 of our revised manuscript.

- We also performed the requested experiments with the ALIX Δ PRR truncation mutant, with the autoinhibitory C-terminal proline-rich region (PRR) deleted. This construct has been shown to increase exosome formation [Larios, et al. (2020) *J. Cell Biol.* 219: e201904113]. However, this construct was only expressed at low levels compared to the only-YFP control and TIRF microscopy showed that it did not locate to the plasma membrane (new Supplementary Fig. 3G). It has previously been shown that this mutant is not recruited to the plasma membrane, likely because binding to lysobisphosphatidic acid is disrupted [Larios, et al. (2020) *J. Cell Biol.* 219: e201904113]. We discuss these new data at page 4–5 of our revised manuscript.

2.5 *I was wondering if overexpression of actin-independent integrins could increase the number/size of ESCRT-III structures. As a control, overexpression of integrins that are linked to actin should be tested.*

This point is not entirely clear: to our knowledge all integrins can connect to the actin cytoskeleton. However, we have performed microscopy experiments with overexpression of YFP-tagged integrin β 2 and compared this with unbound YFP. Although integrin β 2-YFP reached the plasma membrane, it did not notably alter the number and size of the ESCRT structures (new Supplementary Figure 4D and below). See also reviewer #1, point #1.3.

Reviewer #3:

In this manuscript, Stempels et al. have shown that ring-shaped ESCRT structures consisting of IST1 and other ESCRT-III proteins are formed around actin-independent integrin clusters at the plasma membrane. The authors claim that these ESCRT structures are required to protect the plasma membrane from damage/rupture at the sites of cell adhesion via the integrin clusters. However, some aspects are still less convincing and requires more experiments to be confirmed.

3.1 *In page 4 line 10: I think it should be supplementary figure 3B.*

We apologize for this mistake and corrected it.

3.2 *In Fig 2C: It is necessary to stain for integrins in the temporal artery biopsy of GCA. This would further correlate the migratory phenotype of CD68+ and vimentin+ fibroblasts to the presence of integrin clusters at the plasma membrane. In addition, it adds to the manuscript an observation related to a pathological state (e.g., inflammation, cancer, etc..).*

We have performed the requested experiments and performed staining of the temporal artery biopsies for integrin $\beta 2$ (CD18) together with IST and CD68 or vimentin. The data show that both the integrin $\beta 2+$ CD68+ macrophages and integrin $\beta 2+$ vimentin+ fibroblasts express IST1 (new main Figure 7B). These new data are discussed at page 6 of our revised manuscript.

3.3 *In Fig 3: A co-immunoprecipitation experiment should also be performed to show: (i) the potential interaction between IST1 and integrins, (ii) The absence of actin in these integrin clusters surrounded by IST1. This could be obtained by immunoprecipitating IST1 and blotting for integrins and F-actin.*

- We have performed the requested co-immunoprecipitation, but could not detect integrin $\beta 2$ upon pulldown of IST1 (new Supplementary Figure 4B). Please note that we do not expect direct interaction, as the ring and worm-shaped ESCRT structures do not directly colocalize but surround the integrin clusters (main Figure 7A), and the distance is quite far (>200 nm, the diffraction-limited resolution of our microscopy). These new data are discussed at page 5–6 of our revised manuscript.
- We have performed new mass spectrometry data showing the reduced attachment of consensus adhesion proteins [Horton, *et al.* (2015) *Nat. Cell Biol.* 17:1577] to the cell support in response to latrunculin B treatment (new main Figure 8A and new Supplementary Figure 5B). As we found that the number of adherent cells increased almost two-fold with latrunculin B (Figure 8B), this supports our conclusion of actin-independent cell adhesions. These new data are discussed at page 6–7 of our revised manuscript.

3.4 *The authors have downregulated IST1 and Tsg101 expression by siRNA. It is well known that the depletion of Tsg101 disrupts receptor trafficking and increases receptor density at the plasma membrane. It would be also important to measure the total integrin receptor levels by immuno-blotting and the migration rate of adherent cells (e.g., HEK 293 cells). In addition, it is necessary to find out how the integrin clusters are affected by IST1 depletion.*

- We thank the reviewer for alerting us to this, and have measured surface levels of integrin $\beta 2$ and integrin αM by flow cytometry (new Supplementary Figure 5G). Whereas the levels of integrin αM were not altered upon knockdown of TSG101 and ALIX, the expression of integrin $\beta 2$ was on average ~20% lower. As this experiment was performed with cells in suspension (which thus do not form ESCRT structures), these data show that the ESCRT machinery affects integrin $\beta 2$ surface levels independently of the ESCRT structures. Overall, this was one of the main problems that we faced: ESCRT proteins have multiple cellular functions and it was challenging to assign specific functions to the ESCRT structures, because knockdown/overexpression also affected other cellular functions. However, we believe our new data with membrane-disrupting silica crystals (new main Figure 8D) support a role of the ECSRT structures in membrane repair, as we explain at point #1.3.

- IST1 and TSG101 depletion both lowered the levels of integrin β 2 clusters, see Reviewer #1 comment #1.3 (new Supplementary Figure 5F). However, as mentioned above, this is likely a general trafficking defect of integrin β 2 and not related to the ESCRT structures.
- 3.5 *In Fig 4C: The experimental interpretation is unclear. It is said that in the absence of serum the ESCRT structures are formed. Thus, how it is interpreted that the leakage of membrane impermeable DAPI into the cells increased upon absence of serum. If these ESCRT structures are protecting or aiding in membrane repair, it shouldn't decrease the DAPI leakage?*
- The same question was raised by reviewer #2 and we apologize that this was unclear. Please see at comment #2.3, #1.2 and #1.3 for our explanation.
- 3.6 *It would also be more convincing to treat cells with certain chemotherapeutic drugs or any other chemical compounds known to induce mild ruptures in the plasma membrane and to show whether the recruitment of these ESCRT structures to the plasma membrane is being altered.*

As requested, we have induced ruptures in the plasma membrane. We used silica crystals, which are known to induce membrane damage [Beckwith, et al. 2020 *Nat. Comm.* 11:2270]. Compared to membrane rupture by chemotherapeutic drugs or other chemical compounds, these crystals have the advantage that they induce local membrane rupture at the contact sites with the plasma membrane. This is likely more similar to the locally induced membrane rupture caused by cell adhesion. We observed clear recruitment of IST1-GFP, and endogenous IST1, CHMP4A, CHMP1B and ALIX at the contact areas between the plasma membrane and the silica crystals. These data are shown in new main Figure 8D (also shown above, at Reviewer #1 point #1.3) and discussed in the Results section at page 8 of our revised manuscript.

March 7, 2023

RE: JCB Manuscript #202205130R

Prof. Geert van den Bogaart
University of Groningen
Molecular Immunology
Nijenborgh 9
Groningen 9747AG
Netherlands

Dear Prof. van den Bogaart,

Thank you for submitting your revised manuscript entitled "Giant worm-shaped ESCRT-scaffolds surround actin-independent integrin clusters." We would be happy to publish your paper in JCB pending final revisions necessary to meet our formatting guidelines (see details below). Due to the final text length and number of figures the paper will be published as an Article instead of Report.

A. MANUSCRIPT ORGANIZATION AND FORMATTING:

1) Text limits: Character count for Articles is < 40,000, not including spaces. Count includes title page, abstract, introduction, results, discussion, and acknowledgments. Count does not include materials and methods, figure legends, references, tables, or supplemental legends.

2) Figure formatting: Articles may have up to 10 main text figures. Molecular weight or nucleic acid size markers must be included on all gel electrophoresis. Scale bars must be present on all microscopy images, including inset magnifications. Please add scale bars to Figure S1C and zoom magnifications in 1D, 2C, 3A/B, 5A/D, 6C, 7A/B/E/F, 8B/D, S2A, S3A/C/E/F, & S4C/E.

3) Supplemental figures: Articles typically may have up to 5 supplemental figures and 10 videos. You currently exceed this limit but, in this case, we will be able to give you the extra space. Each figure can span a full single page, multi-page figures are not allowed by JCB formatting. Please consolidate the data currently presented in the supplemental figures as much as possible.

Also, please avoid pairing red and green for images and graphs to ensure legibility for color-blind readers. If red and green are paired for images, please ensure that the particular red and green hues used in micrographs are distinctive with any of the colorblind types. If not, please modify colors accordingly or provide separate images of the individual channels.

4) Statistical analysis: Error bars on graphic representations of numerical data must be clearly described in the figure legend. The number of independent data points (n) represented in a graph must be indicated in the legend. Please, indicate whether 'n' refers to technical or biological replicates (i.e. number of analyzed cells, samples or animals, number of independent experiments).

***** If independent experiments with multiple biological replicates have been performed, we recommend using distribution-reproducibility SuperPlots (please see Lord et al., JCB 2020) to better display the distribution of the entire dataset, and report statistics (such as means, error bars, and P values) that address the reproducibility of the findings.**

Statistical methods should be explained in full in the materials and methods. For figures presenting pooled data the statistical measure should be defined in the figure legends. Please also be sure to indicate the statistical tests used in each of your experiments (both in the figure legend itself and in a separate methods section) as well as the parameters of the test (for example, if you ran a t-test, please indicate if it was one- or two-sided, etc.). Also, if you used parametric tests, please indicate if the data distribution was tested for normality (and if so, how). If not, you must state something to the effect that "Data distribution was assumed to be normal but this was not formally tested."

5) Materials and methods: Should be comprehensive and not simply reference a previous publication for details on how an experiment was performed. Please provide full descriptions (at least in brief) in the text for readers who may not have access to referenced manuscripts. The text should not refer to methods "...as previously described." Please also indicate the type of membrane used for immunoblotting/western blots as well as acquisition and quantification methods.

6) For all cell lines, vectors, constructs/cDNAs, etc. - all genetic material: please include database / vendor ID (e.g., Addgene, ATCC, etc.) or if unavailable, please briefly describe their basic genetic features, even if described in other published work or gifted to you by other investigators (and provide references where appropriate). Please be sure to provide the sequences for all of your oligos: primers, si/shRNA, RNAi, gRNAs, etc. in the materials and methods. You must also indicate in the methods the source, species, and catalog numbers/vendor identifiers (where appropriate) for all of your antibodies, including secondary. If antibodies are not commercial, please add a reference citation if possible.

7) Microscope image acquisition: The following information must be provided about the acquisition and processing of images:

a. Make and model of microscope

b. Type, magnification, and numerical aperture of the objective lenses

c. Temperature

d. Imaging medium

e. Fluorochromes

f. Camera make and model

g. Acquisition software

h. Any software used for image processing subsequent to data acquisition. Please include details and types of operations involved (e.g., type of deconvolution, 3D reconstitutions, surface or volume rendering, gamma adjustments, etc.).

8) References: There is no limit to the number of references cited in a manuscript. References should be cited parenthetically in the text by author and year of publication. Abbreviate the names of journals according to PubMed.

9) Supplemental materials: Tables, like figures, should be provided as individual, editable files. A summary of all supplemental material should appear at the end of the Materials and methods section. Please include one brief sentence per item.

10) Video legends: Should describe what is being shown, the cell type or tissue being viewed (including relevant cell treatments, concentration and duration, or transfection), the imaging method (e.g., time-lapse epifluorescence microscopy), what each color represents, how often frames were collected, the frames/second display rate, and the number of any figure that has related video stills or images.

11) eTOC summary: A ~40-50 word summary that describes the context and significance of the findings for a general readership should be included on the title page. The statement should be written in the present tense and refer to the work in the third person. It should begin with "First author name(s) et al..." to match our preferred style.

13) A separate author contribution section is required following the Acknowledgments in all research manuscripts. All authors should be mentioned and designated by their first and middle initials and full surnames. We encourage use of the CRediT nomenclature (<https://casrai.org/credit/>).

14) ORCID IDs: ORCID IDs are unique identifiers allowing researchers to create a record of their various scholarly contributions in a single place. At resubmission of your final files, please consider providing an ORCID ID for as many contributing authors as possible.

15) Journal of Cell Biology now requires a data availability statement for all research article submissions. These statements will be published in the article directly above the Acknowledgments. The statement should address all data underlying the research presented in the manuscript. Please visit the JCB instructions for authors for guidelines and examples of statements at (<https://rupress.org/jcb/pages/editorial-policies#data-availability-statement>).

16) JCB also requires authors to submit Source Data used to generate figures containing cropped gels and western blots with all revised manuscripts. This Source Data consists of fully uncropped and unprocessed images for each gel/blot displayed in the main and supplemental figures. Since your paper includes cropped gel and/or blot images, please be sure to provide one Source Data file for each figure that contains gels and/or blots along with your revised manuscript files. File names for Source Data figures should be alphanumeric without any spaces or special characters (i.e., SourceDataF#, where F# refers to the associated main figure number or SourceDataFS# for those associated with Supplementary figures). The lanes of the gels/blots should be labeled as they are in the associated figure, the place where cropping was applied should be marked (with a box), and molecular weight/size standards should be labeled wherever possible. Source Data files will be directly linked to specific figures in the published article. Source Data Figures should be provided as individual PDF files (one file per figure). Authors should endeavor to retain a minimum resolution of 300 dpi or pixels per inch. Please review our instructions for export from Photoshop, Illustrator, and PowerPoint here: <https://rupress.org/jcb/pages/submission-guidelines#revised>

B. FINAL FILES:

Thank you for this interesting contribution, we look forward to publishing your paper in Journal of Cell Biology.

Sincerely,

Johanna Ivaska, PhD
Monitoring Editor
Journal of Cell Biology

Dan Simon, PhD
Scientific Editor
Journal of Cell Biology

Reviewer #1 (Comments to the Authors (Required)):

I find the author did a good job to address most of my comments as well as comments from the other referees. I still find that the characterization of the role of these peculiar ESCRT structures in generating extracellular vesicle to locally repair plasma membrane is poor. However, authors only suggest that it is a possibility, which seems fair regarding my previous comment. Yet, observations reported in this paper are very intriguing and I believe it is important to share these findings even though the mechanistic and functional characterization could be improved.

Reviewer #2 (Comments to the Authors (Required)):

The authors have perfectly addressed all my concerns and have performed all experiments. I enthusiastically recommend publication.

Reviewer #3 (Comments to the Authors (Required)):

All comments from the 3 reviewers were properly addressed. However, mechanistically the MS is still not very convincing